# SWE-BENCH: CAN LANGUAGE MODELS RESOLVE REAL-WORLD GITHUB ISSUES?

**Carlos E. Jimenez**[* 1,2]    **John Yang**[* 1,2]    **Alexander Wettig**[1,2]

**Shunyu Yao**[1,2]    **Kexin Pei**[3]    **Ofir Press**[1,2]    **Karthik Narasimhan**[1,2]

[1]Princeton University    [2]Princeton Language and Intelligence    [3]University of Chicago

## ABSTRACT

Language models have outpaced our ability to evaluate them effectively, but for their future development it is essential to study the frontier of their capabilities. We find real-world software engineering to be a rich, sustainable, and challenging testbed for evaluating the next generation of language models. To this end, we introduce SWE-bench, an evaluation framework consisting of 2,294 software engineering problems drawn from real GitHub issues and corresponding pull requests across 12 popular Python repositories. Given a codebase along with a description of an issue to be resolved, a language model is tasked with editing the codebase to address the issue. Resolving issues in SWE-bench frequently requires understanding and coordinating changes across multiple functions, classes, and even files simultaneously, calling for models to interact with execution environments, process extremely long contexts and perform complex reasoning that goes far beyond traditional code generation tasks. Our evaluations show that both state-of-the-art proprietary models and our fine-tuned model SWE-Llama can resolve only the simplest issues. The best-performing model, Claude 2, is able to solve a mere 1.96% of the issues. Advances on SWE-bench represent steps towards LMs that are more practical, intelligent, and autonomous.

## 1 INTRODUCTION

Language models (LMs) are rapidly being deployed in commercial products such as chatbots and coding assistants. At the same time, existing benchmarks have become saturated (Kiela et al., 2021; Ott et al., 2022) and fail to capture the frontier of what state-of-the-art LMs can and cannot do. There is a need for challenging benchmarks that more accurately reflect real-world applications of LMs to help shape their future development and usage (Srivastava et al., 2023).

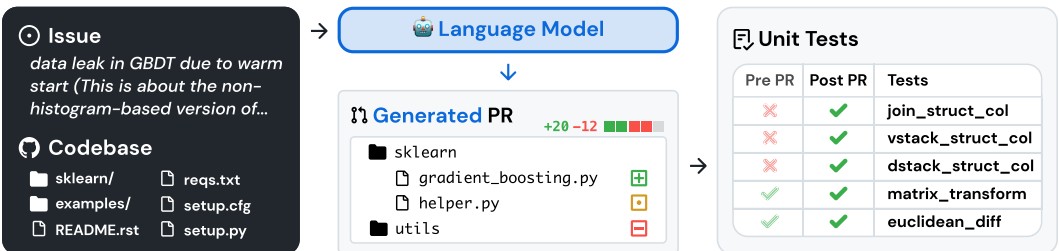

Figure 1: SWE-bench sources task instances from real-world Python repositories by connecting GitHub issues to merged pull request solutions that resolve related tests. Provided with the issue text and a codebase snapshot, models generate a patch that is evaluated against real tests.

Building a good benchmark is difficult since tasks must be challenging enough to stump existing models, but model predictions must also be easy to verify (Martínez-Plumed et al., 2021). Coding

---

*Equal contribution. Correspondence to {`carlosej`, `jy1682`}@princeton.edu.
    Data, code, and leaderboard at swebench.com

tasks are appealing as they pose challenging problems to LMs yet generated solutions can be easily verified by running unit tests. However, existing coding benchmarks, such as HumanEval (Chen et al., 2021), mostly involve self-contained problems that can be solved in a few lines of code.

In the real world, software engineering is not as simple. Fixing a bug might involve navigating a large repository, understanding the interplay between functions in different files, or spotting a small error in convoluted code. Inspired by this, we introduce SWE-bench, a benchmark that evaluates LMs in a realistic software engineering setting. As shown in Figure 1, models are tasked to resolve issues (typically a bug report or a feature request) submitted to popular GitHub repositories. Each task requires generating a patch describing changes to apply to the existing codebase. The revised codebase is then evaluated using the repository's testing framework.

SWE-bench offers several advantages over existing LM programming benchmarks. These include, a realistic setting that utilizes user-submitted issues and solutions, diverse inputs featuring unique code problems from 12 repositories, a robust framework for execution-based evaluation, and the ability to continuously update the benchmark with new instances, requiring minimal human intervention.

We evaluate multiple state-of-the-art LMs on SWE-bench and find that they fail to solve all except the simplest issues. Using a BM25 retriever, Claude 2 is only able to resolve $1.96\%$ of the issues.

In addition to SWE-bench our contributions include the release of a training dataset, SWE-bench-train, which is essential for advancing open model development in this challenging domain. This dataset comprises a collection of 19,000 non-testing task instances derived from 37 repositories. Utilizing SWE-bench-train, we release two fine-tuned models, SWE-Llama 7b and 13b, based on the CodeLlama (Rozière et al., 2023) model. We find that in some settings SWE-Llama 13b is competitive with Claude 2 and is capable of processing contexts exceeding 100,000 tokens.

## 2 SWE-BENCH

SWE-bench is a benchmark featuring GitHub *issues* from popular repositories that report bugs or request new features, and *pull requests* that make changes to the repository to resolve these issues. The task is to generate a pull request that addresses a given issue and passes tests related to the issue.

### 2.1 BENCHMARK CONSTRUCTION

GitHub is a rich data source for software development, but repositories, issues, and pull requests can be noisy, ad-hoc, or poorly documented or maintained. To find high-quality task instances at scale, we use a 3-stage pipeline as follows.

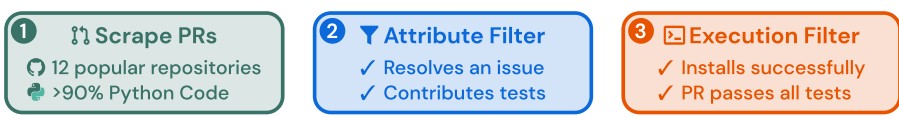

Figure 2: SWE-bench task instances are created from merged pull requests that resolve an issue, contributes tests, and install successfully.

**Stage I: Repo selection and data scraping**. We start by collecting pull requests (PRs) from 12 popular open-source Python repositories on GitHub, producing about $\sim 90{,}000$ PRs in total. We focus on popular repositories as they tend be better maintained, have clear contributor guidelines, and have better test coverage. Each PR has an associated codebase specified by it's base commit.

**Stage II: Attribute-based filtering**. We create candidate tasks by selecting the *merged* PRs that (1) resolve a GitHub issue and (2) make changes to the test files of the repository, which indicates that the user likely contributed tests to check whether the issue has been resolved.

**Stage III: Execution-based filtering**. For each candidate task, we apply the PR's test content, and log the associated test results *before* and *after* the PR's other content is applied. We filter out task instances without at least one test where its status changes from a *fail* to *pass* (henceforth referred to as *fail-to-pass* test). We also filter out instances that result in installation or runtime errors.

Through these stages of filtering, the original 90,000 PRs are filtered down to the 2,294 task instances which comprise SWE-bench. A final breakdown of these task instances across repositories is presented in Figure 3, and Table 1 highlights the key features of SWE-bench task instances. We highlight that the codebases are large with thousands of files, and the reference pull requests often make changes to multiple files at once. Technical details about SWE-bench's construction pipeline are discussed in Appendix A. Additional dataset statistics are in Appendix A.5.

## 2.2 TASK FORMULATION

**Model input.** A model is given an issue text description and a complete codebase. The model is then tasked to make an edit to the codebase to resolve the issue. In practice, we represent edits as patch files, which specify which lines in the codebase to modify in order to resolve the issue.

**Evaluation metrics.** To evaluate a proposed solution, we apply the generated patch, using unix's `patch` program, to the codebase and then execute the unit and system tests associated with the task instance. If the patch applies successfully and all of these tests pass we consider the proposed solution to have successfully resolved the issue. The metric for our benchmark is the percentage of task instances that are resolved. Additional technical details in Appendix A.4.

## 2.3 FEATURES OF SWE-BENCH

Traditional benchmarks in NLP typically involve only short input and output sequences and consider somewhat "contrived" problems created specifically for the benchmark. In contrast, SWE-bench's realistic construction setting imbues the dataset with unique properties, which we discuss below.

**Real-world software engineering tasks**. Since each task instance in SWE-bench consists of a large and complex codebase and a description of a relevant issue, solving SWE-bench requires demonstrating sophisticated skills and knowledge possessed by experienced software engineers but are not commonly evaluated in traditional code generation benchmarks.

**Continually updatable**. Our collection process can be easily applied to any Python repository on GitHub and requires minimal human intervention. Therefore, we can extend SWE-bench with a continual supply of new task instances and evaluate LMs on issues created after their training date, which ensures that the solution was not included in their training corpus.

**Diverse long inputs.** Issue descriptions are typically long and detailed (195 words on average), and codebases regularly contain many thousands of files. Solving SWE-bench requires identifying the relatively small number of lines that need to be edited to solve an issue amongst a sea of context.

**Robust evaluation.** For each task instance, there is at least one *fail-to-pass* test which was used to test the reference solution, and 40% of instances have at least two fail-to-pass tests. These tests evaluate whether the model addressed the problem in the issue. In addition, a median of 51 additional tests run to check whether prior functionality is properly maintained.

**Cross-context code editing.** Unlike prior settings that may constrain edit scope to an individual function or class (e.g., Chen et al., 2021; Cassano et al., 2022) or provide *cloze*-style fill-in blanks (e.g., Lu et al., 2021; Fried et al., 2023), SWE-bench does not provide such explicit guidance. Rather than merely having to produce a short code snippet, our benchmark challenges models to generate revisions in multiple locations of a large codebase. SWE-bench's reference solutions average editing 1.7 files, 3.0 functions, and 32.8 lines (added or removed).

**Wide scope for possible solutions.** The task of repository-scale code editing can serve as a level playing field to compare approaches ranging from retrieval and long-context models to decision-making agents, which could reason and act in code. SWE-bench also allows creative freedom, as models can generate novel solutions that may deviate from the reference PR.

## 3 SWE-LLAMA: FINE-TUNING CODELLAMA FOR SWE-BENCH

It is important to benchmark the performance of open models on SWE-bench alongside proprietary models. At the time of writing, only the CodeLlama models (Rozière et al., 2023) are able to handle the very long contexts necessary. However, we observe that the off-the-shelf CodeLlama variants

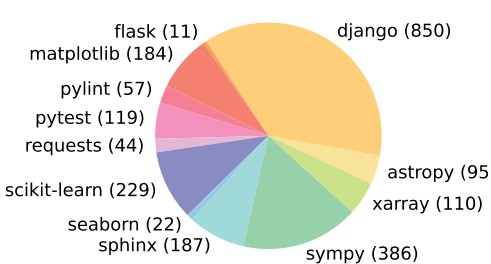

Figure 3: Distribution of SWE-bench tasks (in parenthesis) across 12 open source GitHub repositories that each contains the source code for a popular, widely downloaded PyPI package.

Table 1: Average and maximum numbers characterizing different attributes of a SWE-bench task instance. Statistics are micro-averages calculated without grouping by repository.

|  |  | Mean | Max |
|---|---|---|---|
| Issue Text | Length (Words) | 195.1 | 4477 |
| Codebase | # Files (non-test) | 3,010 | 5,890 |
|  | # Lines (non-test) | 438K | 886K |
| Gold Patch | # Lines edited | 32.8 | 5888 |
|  | # Files edited | 1.7 | 31 |
|  | # Func. edited | 3 | 36 |
| Tests | # Fail to Pass | 9.1 | 1633 |
|  | # Total | 120.8 | 9459 |

are not capable of following the detailed instructions to generate repository-wide code edits, and typically output placeholder responses or unrelated code. To better evaluate the capabilities of these models, we perform supervised fine-tuning on the 7 billion- and 13 billion-parameter CodeLlama-Python models. The resulting models are specialized repository editors that can run on consumer hardware and resolve GitHub issues.

**Training data.** We follow our data collection procedure and collect 19,000 issue-PR pairs from an additional 37 popular Python package repositories. In contrast to Section 2.1, we do not require that pull requests contribute test changes. This allows us to create a much larger training set to use for supervised fine-tuning. To eliminate the risk of data contamination, the set of repositories in the training data is disjoint from those included in the evaluation benchmark.

**Training details.** Given the instructions, an issue text from GitHub and the relevant code files as the prompt, we finetune SWE-Llama to generate the patch that solved the given issue (the "gold patch"). For memory efficiency, we fine-tune only the weights of the attention sublayer using LoRA Hu et al. (2022), and exclude training sequences with more than 30,000 tokens, reducing the effective size of the training corpus to 10,000 instances. More details are provided in Appendix B.

## 4 EXPERIMENTAL SETUP

In this section we explain how inputs are constructed to run SWE-bench evaluation. In addition, we review the models that we evaluate in this work.

### 4.1 RETRIEVAL-BASED APPROACH

SWE-bench instances provide an issue description and a codebase as input to the model. While issues descriptions are usually short (195 words on average as shown in Table 1), codebases consist of many more tokens (438K lines on average) than can typically be fit into an LMs context window. Then the question remains of exactly how to choose the relevant context to provide to the model?

To address this issue for our baselines, we simply use a generic retrieval system to select the files to insert as context. In particular, we evaluate models under two relevant context settings: 1) sparse retrieval and 2) an oracle retrieval.

**Sparse retrieval.** Dense retrieval methods are ill-suited to our setting due to very long key and query lengths, and especially the unusual setting of retrieving code documents with natural language queries. Therefore, we choose to use BM25 retrieval (Robertson et al., 2009) to retrieve relevant files to provide as context for each task instance. We experiment with three different maximum context limits, and simply retrieve as many files as fits within the specified limit. We evaluate each model on all limits that fit within its context window and report the best performance. From observation, models perform best on the shortest context window, as shown in Table 2.

**"Oracle" retrieval.** For analysis purposes we also consider a setting where we "retrieve" the files edited by the reference patch that solved the issue on GitHub. This "oracle" setting is less realistic,

since an engineer working on addressing an issue may not know a priori which files need to be modified. In addition, this setting is also not necessarily comprehensive since edited files alone may not include all the required context to understand exactly how software will behave when interacting with unseen parts of the code.

We compare the BM25 retrieval results with those of the "oracle" retrieval setting, as shown in Table 3. We observe that in approximately 40% of instances, BM25 retrieves a superset of the oracle files for the 27,000-token context limit. However, in almost half of the instances with the 27,000-token limit, it retrieves none of the files from the "oracle" context.

## 4.2 INPUT FORMAT

Once the retrieved files are selected using one of the two methods above, we construct the input to the model consisting of task instructions, the issue text, retrieved files and documentation, and finally an example patch file and prompt for generating the patch file. Examples of instances and further details on this formulation are provided in Appendix D.

## 4.3 MODELS

Due to the need to process long sequence lengths, there are only a few models that are currently suitable for SWE-bench. Thus we evaluate ChatGPT-3.5 (`gpt-3.5-turbo-16k-0613`), GPT-4 (`gpt-4-32k-0613`), Claude 2, and SWE-Llama with their context limits shown in Table 4.

Table 2: Model resolve rates with BM25 retrieval, with different maximum context lengths.

|  | Max. Content | | |
| --- | --- | --- | --- |
| Model | 13k | 27k | 50k |
| Claude 2 | **1.96** | **1.87** | **1.22** |
| SWE-Llama 7b | 0.70 | 0.31 | 0.00 |
| SWE-Llama 13b | 0.70 | 0.48 | 0.00 |

Table 3: BM25 recall with respect to oracle files for different maximum context lengths.

|  | BM25 Recall | | |
| --- | --- | --- | --- |
|  | 13k | 27k | 50k |
| Avg. | 29.58 | 44.41 | 51.06 |
| All | 26.09 | 39.83 | 45.90 |
| Any | 34.77 | 51.27 | 58.38 |

Table 4: We compare the different context lengths and proportion of the "oracle" retrieval setting covered. Models with shorter context lengths are thus inherently disadvantaged. Note that descriptions of token-lengths is a relative non-standard measure (e.g. Llama-tokenized sequences are 42% longer on average than the equivalent sequence tokenized for GPT-4).

|  | ChatGPT-3.5 | GPT-4 | Claude 2 | SWE-Llama |
| --- | --- | --- | --- | --- |
| Max. Tokens | 16,385 | 32,768 | 100,000 | $\geq$100,000 |
| % of Instances | 58.1% | 84.1% | 96.4% | $\geq$94.8% |

## 5 RESULTS

We report results for models using different retrieval mechanisms and prompting styles, then provide some analysis and insight into model performance and difficulty. We summarize models' performance using BM25 retrieval in Table 5. Across the board, models struggle significantly to resolve issues. The best performing model, Claude 2, is only able to resolve 1.96% of the issues.

To analyze the importance of the retriever to the overall system results, we present the "oracle" retrieval results in Appendix Table 18. There, Claude 2 is able to resolve 4.8% of issues using the "oracle" retriever. We further analyze the importance of context in the discussion below.

**Difficulty differs across repositories.** When breaking performance down by repository, all models trend similarly across different repositories as show in Figure 4. Despite this, the issues resolved by each model do not necessarily overlap extensively. For example, in the "oracle" setting Claude 2 and

Table 5: We compare models against each other using the BM25 retriever as described in Section 4. *Due to budget constraints we evaluate GPT-4 on a 25% random subset of SWE-bench.

| Model | % Resolved | % Apply |
|---|---|---|
| Claude 2 | **1.96** | 43.07 |
| ChatGPT-3.5 | 0.17 | 26.33 |
| GPT-4* | 0.00 | 14.83 |
| SWE-Llama 7b | 0.70 | 51.74 |
| SWE-Llama 13b | 0.70 | **53.62** |

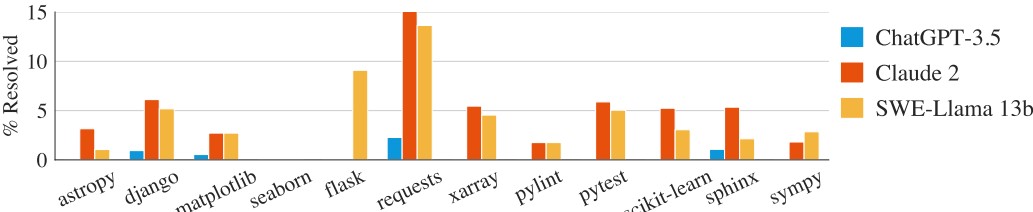

Figure 4: Resolution rate for three models across the 12 repositories represented in SWE-bench in the "Oracle" retrieval setting.

SWE-Llama 13b perform comparably, with each model resolving 110 and 91 instances respectively. Yet of these instances, Claude 2 only solves 42% of the instances solved by SWE-Llama.

This may also be related to the presence of images in issues, which can be encoded into the issue markdown with embedded image links (i.e. `![image][https://...]`). Some repositories naturally feature more instances with images; for example 32% of matplotlib and 10% of seaborn instances contain embedded images in their issue text compared to just 2% of all instances. Solving these instances may require multi-modal LMs or some kind of external tool use to process images.

**Difficulty correlates with context length.** Chat models may be pre-trained on long sequences of code but are typically asked to generate shorter coder snippets with limited context provided to frame the question. As shown in Figure 5, we see that as total context length increases, Claude 2's performance drops considerably; behavior that is also observed in other models. In our evaluation settings, models see a lot of code that may not be directly related to solving the issue at hand, and they seem to frequently struggle with localizing problematic code needing to be updated. This result corroborates other studies showing that models become distracted by additional context and may be sensitive to the relative location of target sequences (Liu et al., 2023b). Even when increasing the maximum context size for BM25 would increase recall with respect to the oracle files, performance drops, as shown in Table 2, as models are simply ineffective at localizing problematic code.

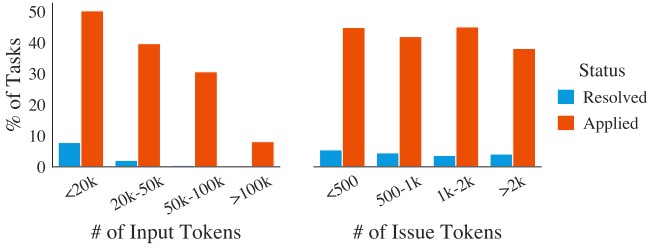

Figure 5: We compare the performance of Claude 2 on tasks partitioned by total input length and by only the issue length.

Table 6: We show the results for the "Oracle"-collapsed retrieval setting, which uses oracle files but collapses code that isn't directly modified by the PR ±15 lines.

| Model | "Oracle"-collapsed | |
|---|---|---|
| | Resolved | Applied |
| ChatGPT-3.5 | 1.09 | 40.93 |
| Claude 2 | **5.93** | **68.18** |
| GPT-4 | 3.40 | 48.65 |

Further investigating this, we provide an input ablation on the "oracle" retrieval context, "oracle"-collapsed, where retrieved files are collapsed entirely, except for the lines actually edited by the true pull request (with ±15 lines of buffer) shown in Table 6. In this setting, we see increases in performance, with GPT-4 jumping from 1.3% to 3.4% and Claude 2 from 4.8% to 5.9%.

**Difficulty does not correlate with issue resolution date.** In Table 7 we show model results in the "oracle" retrieval setting, partitioned by date, for PRs created before or after 2023. We find that for most models there's little difference in performance before or after this date, with the exception of GPT-4. We consider this result to be largely promising as it suggests that despite models having been exposed to some version of an repository's codebase, they are unlikely to "cheat" to address issues simply by generating a more recent version of the repository.

Table 7: We compare performance on task instances from before and after 2023 in the "Oracle" retrieval setting. Most models show little difference in performance. *Due to budget constraints, GPT-4 is evaluated on a 25% random subset of SWE-bench tasks, which may impact performance.

|  | Claude 2 | ChatGPT-3.5 | GPT-4* | SWE-Llama 7b | SWE-Llama 13b |
|---|---|---|---|---|---|
| Before 2023 | **4.87** | 0.49 | **1.96** | 2.95 | **3.98** |
| After 2023 | 4.23 | **0.77** | 0.0 | **3.46** | 3.85 |

**Finetuned models are sensitive to context distribution shifts.** The finetuned models SWE-Llama 7b and 13b perform surprisingly poorly with BM25 retrieved context. As these models were finetuned using the "oracle" retrieval as context, we suspect this shift in context makes it difficult for the model to perform reliably. For instance, SWE-Llama was trained to edit every file included as context whereas in the BM25 setting many files provided in context are not expected to be changed.

**Generating patches is easier than generating whole files.** Models are often trained using standard code files and likely rarely see patch files. We generally formulate our task to have models generate patch files as opposed to recreating the entire file with their proposed change, since patch files will usually be a much more efficient representation of a file change. As shown in Table 5, we observe that models still struggle with generating well-formatted patch files. So we experiment with asking models to instead regenerate entire files with their proposed changes to resolve the issue. In this setting, we find that models generally perform worse at this task than when generating patch files; for instance, Claude 2 scores at 2.2% compared to 4.8% in the main table for "oracle" retrieval. Even when controlling for instance length, generating on the shorter half of the task instances by input tokens yields 3.9% compared to 7.8% for generating patches with Claude 2.

**Language models tend to generate shorter, simpler edits.** Model generated patch files tend to add and remove fewer lines than their respective gold patch. As shown in Table 8, compared to an average gold patch, model generated patch files that apply correctly are less than half the total length (74.5 versus 30.1 lines) of gold edit patch files, and rarely edit more than a single file.

Table 8: Average edits of model generated patches in the "oracle" retrieval setting across successfully applied patches. For the task instances specific to each model, we calculate the same statistics across the gold patches. Avg Gold shows statistics macro-averaged over each models' respective gold patches. All Gold shows statistics for all gold patches unconditioned on model performance.

| Model | Total Lines | Added | Removed | Functions | Files |
|---|---|---|---|---|---|
| Claude 2 | 19.6 | 4.2 | 1.9 | 1.1 | 1.0 |
| Gold | 44.1 | 12.0 | 5.8 | 2.1 | 1.2 |
| ChatGPT-3.5 | 30.1 | 3.8 | 2.7 | 1.6 | 1.0 |
| Gold | 39.6 | 9.5 | 6.1 | 1.9 | 1.2 |
| GPT-4 | 20.9 | 4.4 | 1.5 | 1.0 | 1.0 |
| Gold | 33.6 | 8.4 | 3.8 | 1.9 | 1.1 |
| SWE-Llama 13b | 17.6 | 1.6 | 1.2 | 1.2 | 1.1 |
| Gold | 37.8 | 10.0 | 4.4 | 1.9 | 1.1 |
| SWE-Llama 7b | 16.7 | 1.3 | 1.2 | 1.2 | 1.1 |
| Gold | 40.2 | 11.3 | 4.9 | 1.9 | 1.1 |
| Avg Gold | 39.1 | 10.2 | 5.0 | 1.9 | 1.1 |
| All Gold | 74.5 | 22.3 | 10.5 | 3.0 | 1.7 |

## 5.1 A QUALITATIVE ANALYSIS OF SWE-LLAMA GENERATIONS

We select 11 generations from SWE-Llama and Claude 2 to better understand the quality of the task and generated patches under the "oracle" retrieval setting. Here we discuss an example from SWE-Llama and our overall findings, with in-depth analyses for other examples shown in Appendix F.

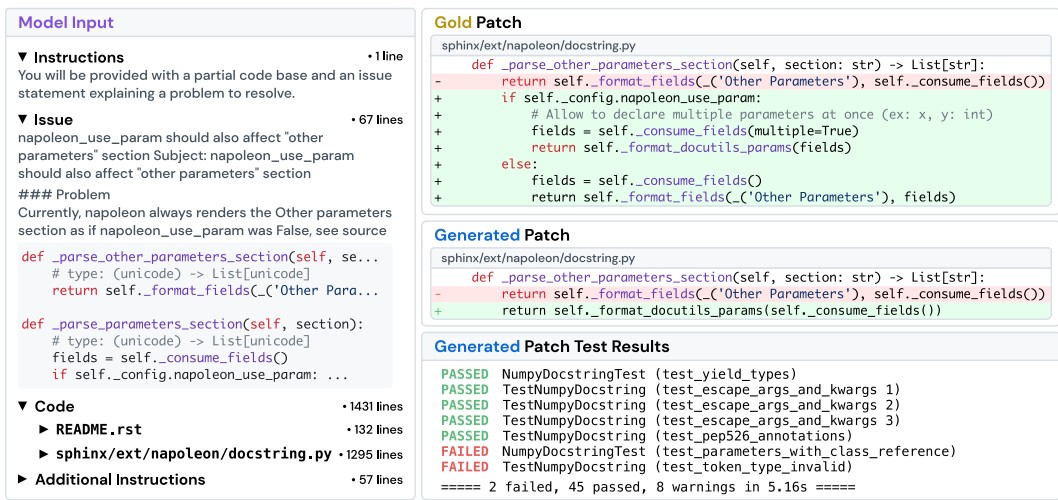

Figure 6: We show an example of an formatted task instance, a model prediction, and the testing framework logs. In the patches, red highlights are deletions. Green highlights are additions.

We'll consider the task instance `sphinx-doc_sphinx-8713` from the Sphinx documentation generator, shown in Figure 6. The issue states that the `napoleon` extension of Sphinx is not properly formatting the documentation keyword "Other Parameters" when the config setting `napoleon.use_param` is set to `True`. The issue text further provides a detailed code snippet of where the problematic source code is suspected to be, as well as some code examples for reproducing the error and additional information related to package versions. For this particular instance, the model did not resolve the task, failing to pass some of the tests resolved by the gold solution.

In the "oracle" retrieval setting, the model input provides this issue text along with some instructions, the full contents of files edited by the gold patch, and an example of the diff format we expect the answer to be in. The total model input consists of 1,558 lines of context or 20,882 tokens. When comparing the gold patch and the model's patch, we find an obvious mistake. While the model edits the correct function, `_parse_other_parameters_section` at line 684 in `sphinx/ext/napoleon/docstring.py`, it changes the function to behave as if `napoleon.use_param` were always `True` instead of checking the config setting first and copying what the `_parse_parameters_section` does, like the gold patch. In the tests, `test_parameters_with_class_reference` directly compares the documentation produced using a config where `napoleon_use_param` is set to `False`, which catches the model's error immediately.

Comparing results across all the examples we consider, we notice a few prominent trends in behavior. Models tend to write primitive Python code and do not leverage existing third-party libraries or the rest of the codebase for their solutions. Models' generations also reflect a "greedy" approach of solving the problem *exactly*, with little regard for code style or logical constraints that might be reflected by the codebase (i.e. using relative instead of absolute imports). In contrast, we observe that many gold patches will make structural improvements that cover a much larger scope of the codebase; these edits not only resolve the issue, but also anticipate and solve potential future issues.

## 6 RELATED WORK

**Evaluation of LMs.** Several recent works for evaluating LMs have either proposed a collection of mutually distinct tasks spanning across multiple domains (Hendrycks et al., 2021; Liang et al., 2022; Srivastava et al., 2023) or turned to the web as an interactive setting featuring tasks that require

multiple steps to solve (Yao et al., 2022; Zhou et al., 2023; Deng et al., 2023; Liu et al., 2023d). There are several drawbacks with such a "potpourri" style setup. First, each task tends to narrowly focus on one or a few skills, resulting in challenges that are typically too simple, pigeonhole the model into a reduced role, and do not provide models with the bandwidth to exercise their versatility or potentially demonstrate new abilities (Srivastava et al., 2023). Consequently, a model's performance on such task conglomerations may not yield actionable, deep insights regarding its capabilities and how to improve them (Schlangen, 2019; Martínez-Plumed et al., 2021; Bowman & Dahl, 2021). SWE-bench addresses these shortcomings, as our work demonstrates that it is significantly challenging, presents a wide range of possibilities for improving LMs to solve this task, and is easy to refresh over time with new task instances, each of which introduce novel, nuanced, and practical challenges.

**Code Generation Benchmarks.** HumanEval (Chen et al., 2021) is the current standard in a long-standing pursuit of synthesizing code from natural language descriptions (Yu et al., 2018; Austin et al., 2021; Hendrycks et al., 2021; Li et al., 2022a; Zan et al., 2023). In the past year, subsequent benchmarks have sought to augment HumanEval with extensions to different languages (Cassano et al., 2022; Athiwaratkun et al., 2023; Orlanski et al., 2023), variations in edit scope (Yu et al., 2023; Du et al., 2023), similar but novel code completion tasks (Muennighoff et al., 2023), and more testing (Liu et al., 2023a). Simultaneously, separate works have sought to introduce new coding paradigms (Yin et al., 2022; Yang et al., 2023) or design library-specific problems (Lai et al., 2022; Zan et al., 2022). Instead of partitioning problems into siloed datasets and curtailing them for simplicity's sake, SWE-bench's collection procedure transforms the source code with minimal post-processing, preserving a much broader set of challenges grounded in real-world software engineering beyond closed form completion, such as patch generation, reasoning over long contexts, navigating a codebase directory, and capturing dependency-based relationships across modules.

**ML for Software Engineering.** To overcome traditional program analysis techniques that may not scale or incorporate natural language, one direction of current software engineering research is to use neural networks, including LMs, to automate real-world software development processes (Maniatis et al., 2023; Zheng et al., 2023; Hou et al., 2023). Use cases include automating commit generation (Jung, 2021; Liu et al., 2023c), PR review (Yang et al., 2016; Li et al., 2022b; Tufano et al., 2021), bug localization Kim et al. (2019); Chakraborty et al. (2018), testing (Kang et al., 2023; Xia et al., 2023; Wang et al., 2023), and program repair (Gupta et al., 2017; Allamanis et al., 2017; Monperrus, 2018; Jiang et al., 2018; Goues et al., 2019; Gao et al., 2022; Dinh et al., 2023; Motwani & Brun, 2023). Most relevant to SWE-bench are works that have sought to apply LMs towards automated program repair (Xia & Zhang, 2022; 2023; Fan et al., 2023; Sobania et al., 2023), guiding code editing with commits (Chakraborty & Ray, 2021; Zhang et al., 2022; Fakhoury et al., 2023). However, none of the existing datasets (Just et al., 2014; Karampatsis & Sutton, 2019) present code context at the scale of SWE-bench. Moreover, SWE-bench can be easily extended to new programming languages and repositories, and it provides a significantly more realistic and challenging arena to carry out experiments towards augmenting LMs with software engineering tools and practices.

# 7 DISCUSSION

**Limitations and future directions.** SWE-bench task instances are all in Python; we hope to apply SWE-bench's task instance collection procedure to expand its coverage to more programming languages and domains. Second, our experiments aim to establish a baseline of the simplest and most straight-forward approaches for this task; we do not intend to constrain future methodologies to the same type of approach and encourage future work to investigate different methods (e.g., agent-based approaches, tool augmented LMs). Lastly, while this work evaluates models using execution-based code testing, relying solely on this method is insufficient to guarantee reliable performance of model generations, as we find automated code generations from LMs can frequently be less comprehensive, efficient, or readable compared to human-written solutions.

**Conclusion.** The complexity of real-world software development processes extends far beyond just code completion. By drawing on the open-source collaborative pipeline, SWE-bench creates a faithful mirror of real world coding environments. This more realistic environment encourages creative solutions that can have immediate applicability in open-source software development. We hope that this benchmark and our other contributions can serve as valuable assets in the future development of LMs that are more practical, intelligent, and autonomous.

## 8 ETHICS STATEMENT

SWE-bench is collected entirely from public repositories with licenses that permit software usage that our contributions are in accordance with. Details of the licenses are included in Table 12. During the collection or evaluation processes, we do not collect information about GitHub users, and the SWE-bench task instances do not use GitHub data beyond what is offered via the public API and website. Our contributions do not involve any human subject participation; we do not perform crowdsourcing or recruit human task workers for any part of SWE-bench, including its collection and evaluation procedures along with the experiments. SWE-bench's filtering criteria for GitHub repositories based on popularity does not implicitly or explicitly rely on any discriminative or biased heuristics for repository selection. For the dataset release, we plan to open source the SWE-bench task instances, the collection and evaluation infrastructure, the experimental results, the training data used for fine-tuning SWE-Llama models, and the SWE-Llama model weights. Following best practice precedents, we will also put forth ample documentation to describe each component and its use, and we will also put in place convenient communication channels for soliciting feedback to improve SWE-bench. SWE-bench does not put forth any immediately harmful insights. We briefly discuss the potential impact of SWE-bench's usage in Section E.

## 9 REPRODUCIBILITY STATEMENT

For our submission, we have uploaded the entirety of the source code as a zipped file that has been properly anonymized. We have organized the codebase such that separate directories correspond to different contributions within the main paper (i.e. dataset collection, evaluation, open source model inference, SWE-Llama training, etc.). The source code contains inline documentation that details purpose and usage of different parts of the codebase. In addition, we also include the full set of 2294 SWE-bench task instances that contains all the components discussed in the main paper. Beyond the documentation in the source code, we include thorough technical details for the collection pipeline and evaluation procedures in Section A.2 and Section A.4 that complements the original details in Section 2 of the main paper. These sections fully cover the logic presented in the code and can be helpful for understanding it. Moving forward, as discussed in the ethics statement, we plan to more formally release SWE-bench to the public as an open source repository with thorough details that describes the benchmark, outlines the code, and details its usage. A major component of SWE-bench is the collection framework, which will be part of the open sourced code. Because of its easily maintainable design, as discussed in the main paper, our hope and belief is that SWE-bench should be highly reproducible.

## 10 ACKNOWLEDGEMENTS

We thank Danqi Chen, Tri Dao, Zexuan Zhong, Tianyu Gao, Will Merrill, Mengzhou Xia, Dan Friedman, Adithya Bhaskar, Austin Watkins, Aatmik Gupta, and Richard Zhu for their valuable feedback and advice. We acknowledge support from the National Science Foundation under Grant No. 2239363 and an Oracle Collaborative Research award. Any opinions, findings, conclusions, or recommendations expressed in this material are those of the author(s) and do not necessarily reflect the views of the National Science Foundation.

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

APPENDIX

In the appendix, we provide more thorough details regarding the dataset construction process, evaluation pipeline, and characterization of the SWE-bench benchmark.

## A    BENCHMARK DETAILS

This section complements Section 2 with a more technical and fine-grained summary of the data collection, execution-based validation, and evaluation procedures, along with a fuller characterization of the task instances.

### A.1    HIGH LEVEL OVERVIEW

**Pull request scraping.** From a list of the top 5,000 most downloaded PyPI libraries during August 2023, we select the top 100 packages, identify each library's corresponding open-source GitHub repository, verify which packages have licenses allowing for free software use, and collect all PRs for these repositories via the GitHub developer API. We elect to source problems from well-trafficked repositories because widespread use usually suggests that the repository has extensive documentation, structured open-source development guidelines, and working, well-formatted code.

**Task instance construction.** We construct candidate task instances from PRs that satisfy three conditions. First, the PR's status must be *Merged*. A *Merged* status indicates that the PR's associated code changes were accepted and incorporated into its parent repository. Second, the PR resolves one or more *issues* in its repository. An issue is defined according to its canonical usage in GitHub as a digital ticket for tracking bugs, enhancements, or any general development goals for a software project. We scan a PR's title, body, and commit messages for linked issues (i.e. "fixes #24"). Third, the PR must introduce one or more new tests. A new test is counted when a PR's code changes edits a file path containing a testing-related keyword (e.g. "test", "testing").

A PR that satisfies these criteria is then converted into a candidate task instance such as the example in Figure 7. The codebase $C$ is identified by the repository's owner/name moniker and the pull request's base commit. Recovering the actual codebase from this information is straightforward. We create mirrors of the original GitHub repositories, where each mirror is uniquely identified as `owner_name`. Cloning a repository's corresponding mirror and checking out the base commit yields $C$ in its pre-PR state. The problem statement $P$ is an aggregate of all related issues' titles and descriptions along with any subsequent comments written before the timestamp of the PR's initial commit to avoid leakage of solution details. A PR's code changes are separated into a test patch and a gold patch $\delta$. $T$ consists of all tests from files edited in the test patch. As shown in Figure 7, both $T$ and $\delta$ are stored as `patch` files. Further details about parsing PR and semantic data is in Appendix A.2.

**Execution-based validation.** We verify the usability of a task instance via execution. For each candidate, we first define a virtual environment to serve as an execution context, then install $C$ before applying any patches, and finally run $T$ once before and once after the solution $\delta$ is applied. A candidate is removed from consideration for the final dataset if any step in the verification process fails. In addition, to ensure that a solution $\delta$ is non-trivial, we compare the pre-solution and post-solution validation logs to check for whether there are one or more tests in $T$ where the status changes from *fail* to *pass*. Lastly, we exclude task instances with tests that invoke newly created functions or classes first introduced in the solution $\delta$. Since naming such constructs is typically an arbitrary process and usually not explicitly specified in the problem statement, resolving tests such as these may be an impossible task even for human developers. Information about execution contexts, codebase installation, determining test statuses from logs, and more are in Appendix A.3.

**Continuous Updates.** SWE-bench's collection process is easily extensible to any open source code repositories, allowing for easy and low-maintenance extension to new programming languages and code domains. This design also provides SWE-bench with temporal robustness; as new language models trained on more recent source code are released over time, SWE-bench can simply be updated to produce new task instances based on PRs created after any LM's training date.

---

https://hugovk.github.io/top-pypi-packages/

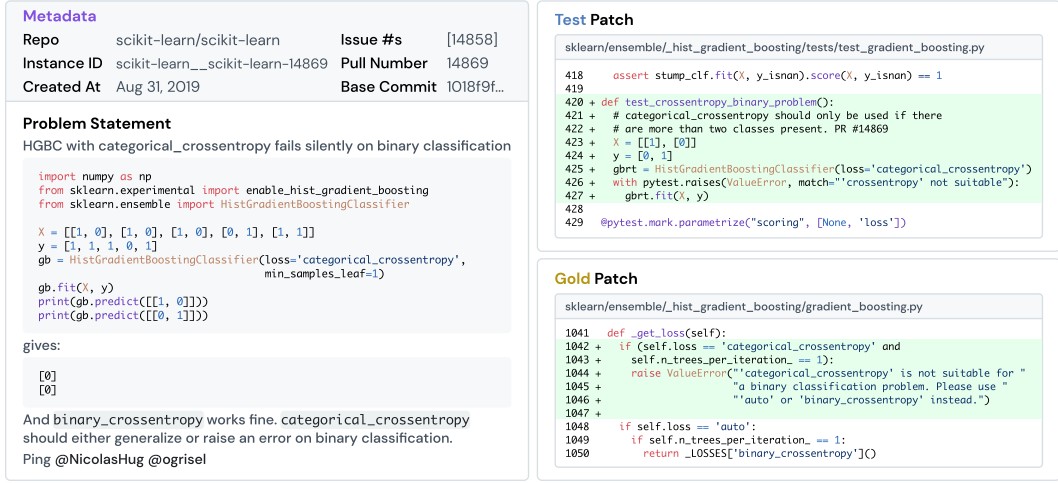

Figure 7: SWE-bench task instance example. Problem statement $P$ is an aggregation of the issues related to the pull request. Codebase $C$ corresponds to a repository and base commit. The tests $T$ and solution $D$ are derived from the original PR's associated code changes. Stylized for readability.

## A.2 CONSTRUCTION PROCESS

We discuss additional details regarding the conversion of a pull request object into a candidate task instance. At a high level, the main goal of this conversion is to acquire relevant information for putting together the codebase $C$, problem statement $P$, unit tests $T$, and solution $\delta$ components introduced in Section 2. To this end, a SWE-bench task instance consists of the following fields, presented in the following Table 9. Collectively, the fields correspond to the four task instance modules.

| Field | Description |
|---|---|
| base_commit | (str) The commit ID that the original PR is applied on top of |
| created_at | (date) Datetime object of when PR was first created (not merged) |
| hints_text | (str) Natural language suggestions for how to solve problem |
| instance_id | (str) A unique identifier created from repo and pull_number |
| issue_numbers | (list) List of issue numbers that the original pull request resolves |
| patch | (str) .patch-format styled string that is a reference solution to the problem, extracted from the original PR's code changes |
| problem_statement | (str) Natural language description of desired change to codebase |
| pull_number | (int) The pull number of the original pull request |
| test_patch | (str) .patch-format styled string containing unseen tests for checking if a task was solved, extracted from the original PR's code changes |
| version | (str) Release version (w.r.t. repo) during which PR was created |
| repo | (str) The repository the task instance originates from |
| FAIL_TO_PASS | (list) List of tests that change in status from *fail* to *pass* |
| PASS_TO_PASS | (list) List of tests that change in status from *pass* to *pass* |
| env_install_commit | (str) Base commit at which to install necessary dependencies for running task instance. |

Table 9: Description of each field of a SWE-bench task instance object. See § A.2 for details regarding how each field is collected.

**Problem Statement.** The problem statement $P$ for each task instance is readily available as the problem_statement field. The problem statement is an aggregate of all issues' first comments along with any comments attached to those issues that were created before the creation date of the PR's initial commit. We crawl for issues from PR's title, body, and commit messages. After

concatenating these components' text data, we first remove any Markdown-style comments, then look through the remaining text for references to issue numbers (a pound # sign followed by a number) and check whether the word preceding the issue number reference is included in a set of keywords suggesting that the issue was resolved by the PR (e.g. "closes", "fixes", "resolves"). The found issues are recorded in the `issue_numbers` field, then separate web requests are made to retrieve each issue's data. To form the `problem_statement`, each issue's title and body are added together and then concatenated with the next issue's if there are multiple. It is also during this step that the `hints_text` field is created and collected from the PR's comment section, where text from comments created before the PR's initial commit. The intuition for this collection methodology is that such PR comments would likely contain natural language and pseudo-code suggestions to the original human task worker regarding how to complete the problem at hand. The experiments presented in this work do not make use of `hints_text`, but we believe this information may be interesting for future investigations.

**Codebase.** The codebase $C$ content is *not* stored in plaintext for every task instance. Rather, the task instance contains a reference to the relevant codebase via the `repo` and `base_commit` field. Both fields are available in the original PR's data. To make retrieval of the codebase $C$ from these two elements reproducible and reliable, we create mirrors of the original repository. Mirrors for the repository constituting both the evaluation and fine tuning data are collected and open-sourced under the SWE-bench GitHub organization. Because an original repository's code may be subject to changes in its commit and edit history outside of the authors' control, we choose to create a mirror repository to ensure that later modifications to the codebase do not potentially render a task instance unusable due to a corruption or removal of the associated `base_commit`. Additionally, we create a mirror instead of cloning and storing the latest version of a repository. This is because a mirror retains the original commit hashes, history, branches, and tags, serving as a faithful and complete history of the technical details of the original repository. A mirror does not retain stars, watchers, issues, or pull requests from the original repository.

We create a mirror from a repository after and within the same day when task instances were collected. The mirror retains the original repository's "`owner/name`" moniker, except that the "/" character is converted to a "_" to confirm to GitHub naming conventions. Given this infrastructure, retrieving a task instance's codebase is straightforward. First, the correct mirror can be cloned from the SWE-bench organization using `repo`. Next, within the local copy of the mirror, checking out the `base_commit` will reset the repository to codebase $C$. To proceed to another task instance from the same repository, `git` version control is used to automatically remove any modifications associated with the current task instance before checking out the next task instance's base commit.

**Solution, Test Patches.** The solution $\delta$ and tests $T$ are derived from the file changes data, or `diff`, of a PR. As mentioned in Section 2.1, the original `diff` along with solution $\delta$ and tests $T$ are represented as a `.patch` file, a format for efficiently specifying transformations to line-based text files. Generally speaking, a `.patch` is structured as a list of blocks, where each block consists of a header and one or more hunks that collectively correspond to changes to a single file. The header contains metadata specifying a file path and line numbers, while the actual modifications to the target file are encoded as multiple lines prefixed by "+" and "-" to indicate additions and removals. To create the tests $T$, we first identifying every unique block within the patch, then pick out and conglomerate blocks with file paths that contain testing-related keywords (e.g. "tests", "testing"). The remaining blocks are merged to form the solution $\delta$. We validate the robustness of the script written to parse correctly $T$ and $\delta$ by applying both patches to the corresponding codebase $C$ and running the tests; we then check that the results reproduce the behavior of the base PR's `diff` data. The solution $\delta$ is saved as the `patch` field while the tests $T$ are saved as the `test_patch` field.

**Remaining Fields.** The `created_at` field is a timestamp that specifies when the base PR was created. We retain the `created_at` field from the original data and use this field to perform temporal analysis of model performance. The `version` field is a string that corresponds to the release version, with respect to the `repo`, during which the PR was released. Depending on availability and the amount of effort required for each method, we create the `version` field by retrieving the information directly from the source code, building the repository locally and invoking code to display the version to standard output, or comparing the `created_at` field with a timeline of release versions

---

Documentation for creating a mirror repository using GitHub

from a repository's webpage. We create executable contexts for every `version` of a repository, as discussed in greater detail in § A.3.

### A.3  EXECUTION-BASED VALIDATION

After filtering through all the PRs from a repository and converting those that satisfy the aforementioned criteria into candidate task instances, the next step is to validate the usability of each task instance via execution. This procedure is broken down into three steps. First, we create executable contexts for each release version of a repository. Next, we check whether the solution $\delta$ and tests $T$ can be applied, installed, and run successfully on top of codebase $C$. Finally, we examine each task instance's execution log to verify a specific set of behaviors to ensure that the task is usable and fair for model evaluation.

**Executable Contexts.** We choose to create executable contexts per release version after experimenting with various degrees of granularity with regards to what definition level to define virtual environments for. Defining task instance-specific contexts is most conducive to ensuring end-to-end installation success, but comes at the cost of laborious manual handcrafting. On the other hand, a repository-specific context based on the latest version of a repository is typically too coarse of a definition that is not compatible with older versions' requirements. We find that release versions are a good proxy for capturing the dependency requirements across a subset of task instances, striking a manageable balance between installation success and manual effort. We manually create each executable context by examining the codebase of the *latest* task instance for each version. Based on the source code and documentation typically found in the repository's `README` and `CONTRIBUTING` guides, we find out the Python version, necessary dependencies, and installation command.

**Validation Engine.** The purpose of the validation engine is to verify candidate task instances. Specifically, this step checks first, that the solution $\delta$ and tests $T$ can be applied to codebase $C$, and second, that the codebase can be properly installed and run within the corresponding virtual environment. To do this, we perform validation repository-by-repository, where for each repository's set of task instances, we perform the following procedure:

1. Create executable contexts as `conda` envs. based on latest task instance per `version`.

2. Group task instances by `version`.

3. Iterate across each task instances group, where for each task instance, we perform the following within the corresponding `conda` env.

   (a) Remove any file changes and checkout the task instance's `base_commit`. This sets the repository to codebase $C$.
   (b) Run the installation command to instantiate codebase $C$.
   (c) Apply the test patch $T$ to codebase $C$.
   (d) Run the testing script, determined from test patch $T$, to generate test result logs $log_{pre}$.
   (e) Apply the solution $\delta$ patch to the codebase $C$ with tests $T$.
   (f) Run the testing script from part (d) again to generate test result logs $log_{post}$.

The testing command consists of the testing framework used by the repository (e.g. `pytest`, `tox`) with paths specified in $T$ appended. The testing command would run any and all tests that are specified within the contents of each file path. If any of the steps $(a)$ through $(f)$ fails, the candidate task instance is discarded from consideration. With moderate variation across repositories, we observe that this step generally removes half of the candidate task instances.

**Examining Validation Logs.** Last but not least, we check the logs $log_{pre}$ and $log_{post}$ created by the validation engine for specific properties. First, to guard against arbitrary naming choices, we check $log_{pre}$ for `ImportError` and `AttributeError` occurrences, which are potentially indicative of dependency naming related errors that would trivial and near-impossible to address correctly. To this end, we remove all task instances with such errors in their $log_{pre}$ from consideration. Next, we compare the test results to check that the task instance is non-trivial, indicated by at least one or more tests having a `fail` status before the solution $\delta$ is applied, then a `pass` status after. To check this, we first define several repository-specific parsers to convert $log_{pre}$ and $log_{post}$ into mappings of test $t_i \in T$ to a status $s \in$ [*fail*,*pass*]. Given these two data structures, we then check that there

| Repo | Total PRs Crawled | Post-Conversion | Post-Validation (Final) |
|---|---|---|---|
| astropy | 9,469 | 1,016 | 95 |
| django | 16,914 | 2,880 | 850 |
| flask | 2,434 | 107 | 11 |
| matplotlib | 16,545 | 1,057 | 184 |
| pylint | 3,848 | 787 | 57 |
| pytest | 5,147 | 750 | 119 |
| requests | 2,344 | 84 | 44 |
| scikit-learn | 15,159 | 1,169 | 229 |
| seaborn | 1,004 | 203 | 22 |
| sphinx | 4,931 | 645 | 187 |
| sympy | 11,928 | 1,897 | 386 |
| xarray | 3,416 | 812 | 110 |
| Total | 93,139 | 11,407 | 2,294 |

Table 10: Statistics for how many candidate task instances were kept after the completion of a stage across the construction and validation procedures.

exists at least one $t_i$ where $s$ changes from *fail* to *pass*. If no such tests are found, the task instance is removed from consideration.

If a task instance fulfills these two criteria, then it is included in the evaluation dataset. Table 10 displays a summary of how many task instances were removed from consideration across the construction process and execution based validation steps. We save all finalized task instances to a single `.json` file that is open sourced and available for download.

Alongside the task instances, we also create a corresponding folder containing the ground truth test results. For each task instance, from their respective $log_{pre}$ and $log_{post}$ test-to-status mappings, we create a test results data structure where the keys are FAIL_TO_FAIL, FAIL_TO_PASS, PASS_TO_FAIL, and PASS_TO_PASS, and the values are lists of tests. By "caching" these results, we remove the need to re-run the solution $\delta$ at evaluation time (although re-running is an available option). We use this data structure to verify task completion, as discussed in Section A.4.

## A.4   EVALUATION PROCEDURE

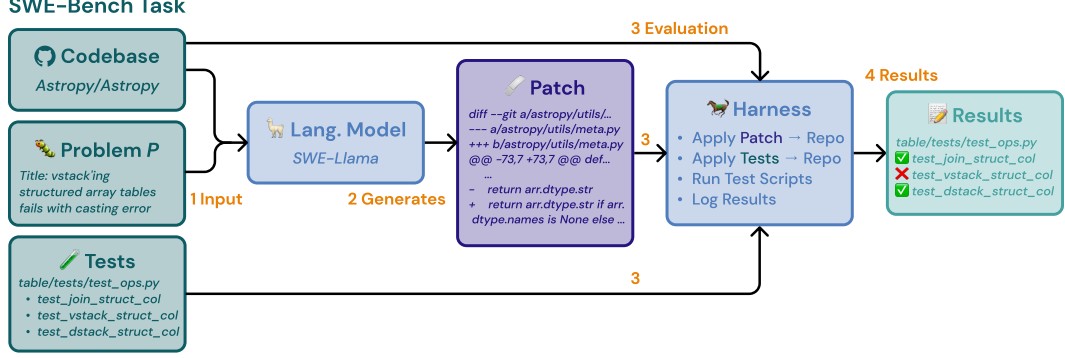

Figure 8: Visualization of the evaluation pipeline at an individual task instance level. During evaluation, the *Patch* is model generated. A prediction `.patch` must be applied successfully and produce the same results as the corresponding task instance's $D$ for task completion.

We provide a visualization of the evaluation procedure in Figure 8. The evaluation procedure scores the model's $\hat{\delta}$ `.patch` generation with respect to the behavior of the solution $\delta$. At a finer-grained level, the evaluation procedure can be broken down into four separate steps, highlighted by the numbered steps in Figure 8. First, the codebase and problem statement are visible and given to the

LM; the LM then generates a `.patch` prediction $\hat{\delta}$. In the evaluation step, the following steps are performed per prediction on the target task instance:

1. Remove any file changes and checkout the task instance's base commit. This sets the repository to codebase $C$.

2. Activate the executable context corresponding to the task instance's `version`.

3. Run installation command to instantiate codebase $C$.

4. Apply test patch $T$ to codebase $C$.

5. Apply prediction patch $\hat{\delta}$ to codebase $C$ with tests $T$.

6. If the previous step fails, we attempt to fix prediction patch $\hat{\delta}$ automatically and reapply it.

7. Run the testing script, determined from test patch $T$, to generate test result logs $log_{\hat{\delta}}$.

Steps 1 through 4 reliably do not fail due to verification during the task instance validation process. If applying the prediction patch (Step 5) fails, we attempt to repair the prediction patch file by removing unnecessary context lines and recalculating the header values (Step 6). If the remaining patch fails again or running the test command (Step 7) fails, then the prediction is automatically given a score of 0. Assuming these steps succeed, the output log $log_{\hat{\delta}}$ can then be converted to a test-to-status mapping, identical in structure to the via the appropriate, repository-specific parser introduced in § A.3.

**Evaluation Metrics Calculation.** To determine task completion, we compare the test-to-status mapping parsed from $log_{\hat{\delta}}$ with the list of tests corresponding to the `FAIL_TO_PASS` and `PASS_TO_PASS` keys from the ground truth test results data structure. Determining task completion is straightforward; we check that all `FAIL_TO_PASS` and `PASS_TO_PASS` tests are found and have a *pass* status in the evaluation test-to-status mapping. If a test is missing or has a non-*pass* status, it is considered a *fail* status. As defined and used in the main paper, a task is considered solved if all tests across `FAIL_TO_PASS` and `PASS_TO_PASS` pass.

## A.5 EVALUATION TEST SET CHARACTERIZATION

We include an expanded form of Table 1 that includes repository specific statistics in Table 11. Table 12 presents a brief description of each repository extracted from the repository's documentation along with the repository's associated open source license. The associated licenses all permit non-commercial usage of the original library source code as long as the permissions in the original licenses are upheld and retained. In addition to the original statistics presented in Table 1, we introduce three new values. The $\delta$ # Lines Added and $\delta$ # Lines Removed together sum up to $\delta$ Lines Edited. "Added" refers to the number of new lines that are introduced, while "Removed" are pre-existing lines taken out by the solution. The $|T|$ (Pass to Pass) statistic refers to the number of tests that were passing before the solution $\delta$ was applied during the validation pipeline. Unlike *fail* to *pass* tests that are intended to characterize the problem statement $P$ and determine if a revision addresses the issue, *pass* to *pass* tests are included to ensure that the revision does not break or violate any existing expected behavior. These tests are extracted during the validation log examination phase as discussed in § A.3. We note that *fail* to *fail* tests and *pass* to *fail* tests are not considered during evaluation, and those statistics are not reflected in the above table.

**Task Instance Issue Categories.** To provide a better sense of the types of problems that SWE-bench task instances include, we perform simple analyses on the issues, identified by the `issue_numbers` field, for each task instance. Per issue, we inspect metadata, specifically tags, to characterize the type of contribution put forth by the PR. Table 13 groups and shows several examples of the 2,289 tags we found across all issues. While the absolute majority of issues are associated with bug fixes, SWE-bench's task instances are associated with a diverse set of code changes with purposes beyond debugging and error correction.

**Attribute Distributions.** In Figure 9, we present plots of the cumulative distribution function for attributes introduced in Table 1. From these plots, we see that the median SWE-bench task instance has a problem description of 140 words, and will take place within a codebase containing just shy of 1900 files and 400K lines. The corresponding reference solution $\delta$ will usually edit a single function

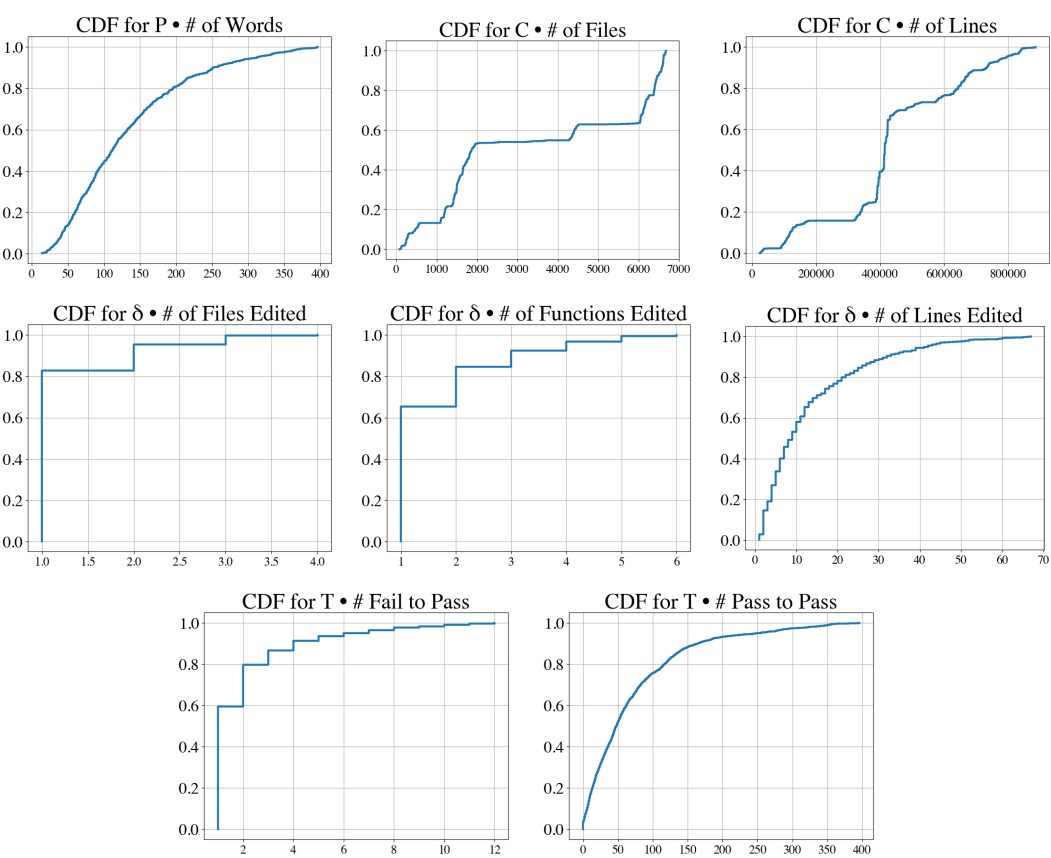

Figure 9: Cumulative Distribution Functions for different attributes of SWE-bench task instances.

| | astropy | django | flask | matplotlib | pylint | pytest |
|---|---|---|---|---|---|---|
| $P$ Length (Characters) | 2,742 | 1,307 | 1,185 | 2,381 | 2,011 | 2,948 |
| $C$ # Files | 1,811 | 6,356 | 225 | 4,395 | 2,426 | 497 |
| $C$ # Lines | 804k | 407k | 35k | 646k | 109k | 111k |
| $\delta$ # Files Edited | 1.5 | 1.5 | 1.6 | 1.5 | 1.8 | 1.4 |
| $\delta$ # Func. Edited | 2.5 | 2.0 | 2.4 | 2.2 | 1.8 | 1.7 |
| $\delta$ # Lines Edited | 36.0 | 18.5 | 35.4 | 58.9 | 36.0 | 24.5 |
| $\delta$ # Lines Added | 25.0 | 12.8 | 23.7 | 35.7 | 26.6 | 18.2 |
| $\delta$ # Lines Removed | 10.9 | 5.7 | 11.6 | 23.2 | 9.5 | 6.4 |
| $|T|$ (Fail to Pass) | 21.7 | 8.8 | 1.4 | 2.4 | 6.8 | 4.1 |
| $|T|$ (Pass to Pass) | 191.0 | 85.9 | 32.5 | 242.4 | 47.0 | 60.7 |
| $|T|$ (All) | 212.8 | 94.6 | 33.9 | 244.8 | 53.7 | 64.8 |

| | requests | scikit-learn | seaborn | sphinx | sympy | xarray |
|---|---|---|---|---|---|---|
| $P$ Length (Characters) | 1,654 | 2,239 | 1,667 | 1,888 | 1,213 | 3,515 |
| $C$ # Files | 119 | 1,224 | 273 | 1,483 | 1,666 | 260 |
| $C$ # Lines | 30k | 361k | 105k | 423k | 678k | 137k |
| $\delta$ # Files Edited | 1.64 | 1.68 | 1.77 | 1.51 | 1.9 | 2.45 |
| $\delta$ # Func. Edited | 1.59 | 2.24 | 1.41 | 2.72 | 3.22 | 3.16 |
| $\delta$ # Lines Edited | 25.5 | 44.0 | 30.1 | 30.6 | 36.3 | 124.8 |
| $\delta$ # Lines Added | 19.2 | 32.7 | 24.9 | 22.0 | 24.2 | 95.6 |
| $\delta$ # Lines Removed | 6.2 | 11.3 | 5.2 | 8.6 | 12.1 | 29.2 |
| $|T|$ (Fail to Pass) | 7.6 | 7.5 | 12.9 | 2.3 | 2.2 | 58.5 |
| $|T|$ (Pass to Pass) | 87.1 | 150.7 | 86.8 | 45.1 | 74.5 | 297.5 |
| $|T|$ (All) | 94.7 | 158.2 | 99.7 | 47.4 | 76.8 | 356.1 |

Table 11: Average numbers characterizing different attributes of a SWE-bench task instance grouped by repository. In addition to the statistics presented in Table 1, we also introduce three new values: $\delta$ # Lines Added, $\delta$ # Lines Removed, and $|T|$ (Pass to Pass).

| Repository | Summary | License |
|---|---|---|
| astropy/astropy | Astronomy and astrophysics core library | BSD 3-Clause |
| django/django | Web framework for building web applications | BSD 3-Clause |
| pallets/flask | Lightweight framework for small web apps | BSD 3-Clause |
| matplotlib/matplotlib | Plotting library for creating visuals | Custom |
| pylint-dev/pylint | Static code analyser for Python 2 or 3 | GPL 2.0 |
| pytest-dev/pytest | Testing framework for Python | MIT |
| psf/requests | Simple, elegant library for writing HTTP requests | Apache-2.0 |
| scikit-learn/scikit-learn | Machine Learning in Python | BSD 3-Clause |
| mwaskom/seaborn | Statistical data visualization in Python | BSD 3-Clause |
| sphinx-doc/sphinx | Library for creating documentation | Custom |
| sympy/sympy | Computer algebra system written in Python | Custom |
| pydata/xarray | N-D labeled arrays and datasets | Apache-2.0 |

Table 12: Summary and licenses for all GitHub repositories that task instances were extracted from.

within a file, changing ∼15 lines, and has a single *fail* to *pass* test to verify the correctness of the change along with 51 *pass* to *pass* tests to check whether existing behavior is preserved.

**Patch Fix Rate.** We present Table 14, which presents summary statistics of how many task instances each model generated patches for (out of 2294), how many of these patches applied successfully, and how many of the successfully applied patches required undergoing the patch fixing procedure introduced in Appendix A.4. We find that fixed patches tend to make up a smaller percentage of the SWE-Llama patches that successfully applied, suggesting that SWE-Llama's fine tuning procedure has a positive effect on generating well-formatted patches. For closed source models, fewer patches apply successfully, and of the ones that do, a greater percentage require the post-generation fix, suggesting that models still struggle with patch generation and structured outputs in general.

| Category | Count | Examples |
|---|---|---|
| Bug | 442 | "Bug" (179); "type:bug" (114); "bug" (57); "type: bug" (48); "Bug :beetle:" (23); "status: confirmed bug" (20);; |
| Feature | 167 | "type:enhancement" (47); "Enhancement" (25); "New feature" (24); "Feature Request" (22); "type: enhancement" (19); "Enhancement :star:" (15); "New Feature" (7); "enhancement" (6); |
| Regression | 39 | "type: regression" (14); "Regression" (14); "regression" (8); |
| Other | 1641 | "help wanted" (71); "good first issue" (66); "printing" (58); "extensions:autodoc" (58); "Easy" (57); "Easy to Fix" (54); "domains:py" (27); "core" (26); "sets" (23); "Wrong Result" (23); "units" (22); "Good first issue" (21); |

Table 13: Categories of tags associated with issues from SWE-bench's task instances.

| Model | Retrieval Setting | Generations | Applies | Fixed | Patch Fix % |
|---|---|---|---|---|---|
| ChatGPT-3.5 | BM25 13k | 2,270 | 604 | 363 | 60.1% |
| ChatGPT-3.5 | "Oracle" | 1,262 | 500 | 222 | 44.4% |
| ChatGPT-3.5 | "Oracle"-collapsed | 1,811 | 939 | 420 | 44.73% |
| Claude 2 | BM25 13k | 2,281 | 988 | 302 | 30.57% |
| Claude 2 | "Oracle" | 2,138 | 1,441 | 360 | 24.98% |
| Claude 2 | "Oracle"-collapsed | 2,242 | 1,564 | 465 | 29.73% |
| GPT-4 | BM25 27k | 573 | 85 | 59 | 69.41% |
| GPT-4 | "Oracle" | 462 | 195 | 121 | 62.05% |
| GPT-4 | "Oracle"-collapsed | 2,292 | 1,116 | 684 | 61.29% |
| SWE-Llama 13b | BM25 13k | 2,010 | 1,230 | 369 | 30.0% |
| SWE-Llama 13b | "Oracle" | 2,125 | 1,532 | 378 | 24.67% |
| SWE-Llama 7b | BM25 13k | 2,139 | 1,187 | 340 | 28.64% |
| SWE-Llama 7b | "Oracle" | 2,119 | 1,503 | 298 | 19.83% |

Table 14: Statistics for how many patches for 2,294 task instances were generated, applied successfully, and required a post-generation fix to apply successfully for each [model, retrieval setting] combination during evaluation. The GPT-4 BM25 27k and "Oracle" settings were ran on the 25% subset. The GPT-4 "Oracle"-collapsed setting was run on the full SWE-bench test set.

## A.6    DEVELOPMENT SET CHARACTERIZATION

In addition to the evaluation test set, we also provide a development set for evaluating models and tuning hyperparameters before running on the final test set. Following the style of tables and graphs from before, we present similar statistics to characterize the 225 development task instances (slightly more than 10% of the main evaluation set) collected from 6 open source repositories with licenses permitting such usage. The development set was collected following the exact same set of methodologies and filters as the main evaluation set. In addition to the pre-existing steps, we also filter the development set to keep task instances that were created after January 1, 2019. Similar to Table 12, in Table 15, we briefly summarize the purpose and licenses of the 6 selected repositories.

Following Table 13, we also list the tags associated with the the development set tasks in Table 16, again showcasing the diversity and coverage of task types beyond fixing bugs. Compared to the main evaluation tasks, we can also see tags (e.g., "Crash :collision:", "io") that refer to issues presenting problems which are unique to the repositories in the development set.

Following Table 1, we present the same set of repository-specific average statistics for the 6 repositories in the development set in Table 17. Across the entire development set, each task instance has 19.9 average / 2 median F2P tests. There are 171.3 average / 79.0 median P2P tests, and 191.2 average / 101.0 median tests in total per task instance.

| Repository | Summary | Count | License |
|---|---|---|---|
| marshmallow-code/ marshmallow | Parse complex objects to/from Python data-types | 9 | MIT |
| pylint-dev/astroid | Library for AST parsing, static analysis/inference | 31 | LGPL-2.1 |
| pydicom/pydicom | Read/modify/write DICOM files w/ Python | 56 | Custom |
| pvlib/pvlib-python | Simulate photovoltaic energy systems performance | 63 | Custom |
| pyvista/pyvista | 3D plotting, mesh analysis through interface | 16 | MIT |
| sqlfluff/sqlfluff | SQL linter, supports multiple dialects, templates | 50 | MIT |

Table 15: Summary and licenses for all GitHub repositories that development task instances were extracted from.

| Category | Count | Examples |
|---|---|---|
| Bug | 127 | "bug" (111); "Bug :cockroach:" (10); "rule bug" (6); |
| Feature | 55 | "enhancement": 46; "Enhancement :star:": 5; "feature-request": 2; |
| Regression | 4 | "Regression" (4); |
| Other | 95 | "api": 11, "documentation": 7, "help wanted": 6, "config options": 5, "io": 5, "jinja": 4, "good first issue": 4, "parser" 3 |

Table 16: Categories of tags associated with issues from SWE-bench's development task instances.

# B  ADDITIONAL DETAILS ON TRAINING SWE-LLAMA

## B.1  TRAINING DETAILS

**Optimization.** We finetune using LoRA (Hu et al., 2022) with $r = 16$, $\alpha = 16$, dropout $= 0.05$, on the query, key, value, and output projection matrices of every attention sublayer. We train with a learning rate of $6e - 4$ and a batch size of 32 sequences per gradient step for a maximum of 4 epochs. During training, we save checkpoints every 50 steps, and after training, select the best checkpoint based on the validation loss on a held-out 100 instances. SWE-Llama 7b was initialized with CodeLlama-Python 7b and trained in 20 hours on 4 NVIDIA A100s. SWE-Llama 13b was initialized with CodeLlama-Python 13b and trained in 47 hours on 8 NVIDIA A100s. We used DeepSpeed Ulysses (Jacobs et al., 2023) and Flash Attention (Dao et al., 2022) to enable long context training.

# C  ADDITIONAL RESULTS

## C.1  RESULTS WITH "ORACLE" RETRIEVAL

Using the "oracle" retrieval method described in Section 4.1, we show the general performance results in Table 18. Naturally, providing only the files edited by the reference solution's pull request, model performance improves compared to the noisier BM25 retrieval setting.

## C.2  EVALUATION TEST SET

We include a repository-by-repository breakdown of model performance in Table 19 that corresponds to Figure 4 in the main paper. As discussed, in the main paper, performance differs heavily across repositories.

## C.3  GPT-4 EVALUATION SUBSET RESULTS

In this section, we present the statistics shown in Table 5 for the 25% random subset that GPT-4 was tested in Table 20. As the selection of the subset is random, we find that the % Resolved and % Apply rates are consistent with the main results, and not significantly skewed towards being simpler or more difficult than the general evaluation set.

| | astroid | marshmallow | pvlib | pydicom | pyvista | sqlfluff |
|---|---|---|---|---|---|---|
| $P$ Length (Characters) | 2199 | 1619 | 1790 | 2076 | 1475 | 2639 |
| $C$ # Files | 252 | 82 | 294 | 455 | 866 | 2297 |
| $C$ # Lines | 60K | 22K | 459K | 170K | 661K | 205K |
| $\delta$ # Files Edited | 2.51 | 1.89 | 1.83 | 1.54 | 2.1 | 3.26 |
| $\delta$ # Func. Edited | 3.03 | 2.11 | 2.89 | 2.23 | 3.0 | 2.71 |
| $\delta$ # Lines Edited | 83.1 | 36.2 | 93.3 | 42.0 | 101.0 | 102.5 |
| $\delta$ # Lines Added | 52.8 | 24.7 | 67.0 | 29.7 | 79.4 | 63.6 |
| $\delta$ # Lines Removed | 30.3 | 11.6 | 26.4 | 12.3 | 21.6 | 38.9 |
| $|T|$ (Fail to Pass) | 23.2 | 53.0 | 19.1 | 24.0 | 8.8 | 14.8 |
| $|T|$ (Pass to Pass) | 182.6 | 242.9 | 107.5 | 176.1 | 96.5 | 239.7 |
| $|T|$ (All) | 205.8 | 295.9 | 126.6 | 200.1 | 105.3 | 254.5 |

Table 17: Average numbers characterizing different attributes of a SWE-bench task instance grouped by repository for repositories in the development dataset. The same statistics presented in Table 11 are also shown here.

| | BM25 Retrieval | | "Oracle" Retrieval | |
|---|---|---|---|---|
| Model | % Resolved | % Apply | % Resolved | % Apply |
| Claude 2 | **1.96** | 43.07 | **4.80** | 62.82 |
| ChatGPT-3.5 | 0.17 | 26.33 | 0.52 | 21.80 |
| GPT-4* | 0.00 | 14.83 | 1.74 | 34.00 |
| SWE-Llama 7b | 0.70 | 51.74 | 3.01 | 65.52 |
| SWE-Llama 13b | 0.70 | **53.62** | 3.97 | **66.78** |

Table 18: We compare models against each other using the BM25 and oracle retrieval settings as described in Section 4. The main results table, Table 5, presents the results for the different models when using BM25 only. *Due to budget constraints we evaluate GPT-4 on a 25% random subset of SWE-bench in the "Oracle" and BM25 27K retriever settings only.

## C.4 EXTENDED TEMPORAL ANALYSIS

In this section, we present an extended temporal analysis of task instances solved by year that follows the analysis shown in Table 7 of the evaluation section in the main paper. In Table 21, we present the % Resolved statistic across models under the "Oracle" retrieval setting for 6 different temporal partitions that group tasks by the years in which the issues were created. It is evident from the table that there is no consistent correlation between model performance and year, supporting our conclusion that despite having potentially seen older versions of code within its pre-training datasets, understanding and implementing in fixes in SWE-bench is a difficult task that requires understanding and cannot be accomplished feasibly or consistently via memoization of observed data.

## C.5 F2P, P2P RATE ANALYSIS

In the main paper results, we present the "% Resolved" statistic that indicates how many task instances were *completely* solved by the different models. In this section, we provide more fine-grained insight into the gap of task instances where 1. The model's patch generation was applied successfully and 2. The task instance was not resolved. Assuming a patch is applied successfully, we define 6 cases in Table 22 that fully capture the distribution of all possible outcomes based on the pass/fail results of F2P and P2P tests. In addition to the "Resolved" outcome that has been established, we introduce five new terms. The "Breaking Resolved" outcome refers to when the desired behavior of the issue has been accomplished (all F2P tests pass), but not all prior behavior is maintained (not all P2P tests pass). "Partially Resolved" refers to when prior behavior of a codebase was maintained (all P2P tests pass); however, the desired behavior is not fully accomplished (not all F2P tests pass). The "Work in Progress" case is when the desired behavior is not fully accomplished (not all F2P tests pass) and the prior behavior of the codebase is not maintained (not all P2P tests pass). A "No-Op" is when a code change does not have any effect on the original codebase; prior

| Repo | Claude 2 | ChatGPT-3.5 | GPT-4 | SWE-Llama 13b | SWE-Llama 7b |
|---|---|---|---|---|---|
| astropy/astropy | 3.23 | 0.00 | 0.00 | 1.06 | 3.16 |
| django/django | 6.15 | 1.32 | 2.50 | 5.19 | 4.00 |
| matplotlib/matplotlib | 3.05 | 3.33 | 0.00 | 3.12 | 1.11 |
| mwaskom/seaborn | 0.00 | 0.00 | 0.00 | 0.00 | 0.00 |
| pallets/flask | 0.00 | 0.00 | 0.00 | 9.09 | 0.00 |
| psf/requests | 15.91 | 2.33 | 8.33 | 13.64 | 18.18 |
| pydata/xarray | 6.90 | 0.00 | 0.00 | 5.81 | 3.00 |
| pylint-dev/pylint | 1.75 | 0.00 | 0.00 | 1.75 | 1.75 |
| pytest-dev/pytest | 5.93 | 0.00 | 0.00 | 5.04 | 4.20 |
| scikit-learn/scikit-learn | 5.41 | 0.00 | 0.00 | 3.12 | 0.88 |
| sphinx-doc/sphinx | 5.65 | 1.83 | 0.00 | 2.25 | 2.69 |
| sympy/sympy | 1.94 | 0.00 | 0.00 | 3.01 | 1.59 |

Table 19: % Resolved for models per repository represented in SWE-bench.

| | BM25 Retrieval | | "Oracle" Retrieval | |
|---|---|---|---|---|
| Model | % Resolved | % Apply | % Resolved | % Apply |
| Claude 2 | 2.27 ↑ 0.31 | 45.72 ↑ 2.65 | 4.01 ↓ 0.79 | 62.65 ↓ 0.17 |
| ChatGPT-3.5 | 0.17 −0.00 | 26.53 ↑ 0.02 | 0.70 ↑ 0.18 | 21.64 ↓ 0.16 |
| GPT-4 | 0.00 −0.00 | 14.82 −0.00 | 1.74 −0.00 | 34.00 −0.00 |
| SWE-Llama 7b | 0.35 ↑ 0.35 | 49.04 ↓ 2.70 | 1.92 ↓ 1.09 | 63.70 ↓ 1.82 |
| SWE-Llama 13b | 0.70 −0.00 | 56.54 ↑ 2.92 | 4.54 ↑ 0.57 | 66.67 ↓ 0.11 |

Table 20: We compare models against each other using the BM25 and oracle retrieval settings as described in Section 4 on a 25% random subset (574 instances) of SWE-bench in the "oracle" and BM25 27K retriever settings only. This is the same subset that GPT-4 is evaluated on, as mentioned in Table 5. The difference relative to percentages in Table 5 and Table 18 is included as a subscript.

behavior is maintained (all P2P tests pass) but the issue remains completely unresolved (0 F2P tests pass). Finally, if the issue is unresolved (0 F2P tests pass) and prior working behavior is reverted (some P2P tests fail), the codebase is left in a worse state, which we define to be a "Regression".

In Table 23, we categorize patch generations that successfully applied according to these six cases. We find that of non-"Resolved" issues, the majority of patch generations proposed by the model do not solve a single F2P test case from the corresponding task instance ("No-Op" and "Regression"). Within the subset of these cases, the majority (60% to 70%) of cases are a No-Op, while the model breaks existing behavior for the remainder of these situations.

Generally, the cases where model generations pass some, but not all tests ("Breaking Resolved", "Partially Resolved", "Work in Progress") cumulatively represent a smaller subset of problems relative to the other three categories. From manual inspection of several of these cases, it is clear that the model demonstrates some understanding of the task requirements. However, due to the baseline methods' limited view of the codebase that does not include information such as inter-file dependencies and functions' relationships, for many of these task instances often fail because a change that correctly resolves the immediate issue does not account for other modules that use and are affected by the changed entity. We include several case studies that directly highlight these shortcomings in Section F Overall, these results highlight not just the difficulty of SWE-bench, but also point to the potential value of providing feedback via an execution environment that would allow models to run fixes against existing tests, then decide whether to continue editing or submit the patch for review.

## C.6 PATCH GENERATION EXTENDED ANALYSIS

In this section, we present a statistics to quantify various facets of patch generations following the metrics laid out in Table 8. In Table 24, we recalculate these values for *all* patch generations in the oracle retrieval setting for all models, regardless of whether or not the patch was applied

| Year | Total | 25% | Claude 2 | GPT-3.5 | GPT-4* | SWE-Llama 13b | SWE-Llama 7b |
|------|-------|-----|----------|---------|--------|---------------|--------------|
| 2023 | 244 | 61 | 4.51 | 1.56 | 0.00 | 4.07 | 3.50 |
| 2022 | 395 | 117 | 4.05 | 0.85 | 3.42 | 2.80 | 2.46 |
| 2021 | 383 | 102 | 4.18 | 0.00 | 2.94 | 4.45 | 2.56 |
| 2020 | 427 | 109 | 5.15 | 0.71 | 0.00 | 3.96 | 3.43 |
| 2019 | 437 | 112 | 5.72 | 1.49 | 1.79 | 4.55 | 2.21 |
| 2018 | 165 | 37 | 5.45 | 0.00 | 0.00 | 3.57 | 2.94 |
| < 2018 | 89 | 36 | 4.49 | 0.00 | 2.78 | 3.37 | 1.09 |

Table 21: We present an extended temporal analysis in this table, showing the % resolved for task instances across models in the "Oracle" retrieval setting, separated by different cutoff dates. The *Year* column refers to the subset of tasks that were created during the specified calendar year. In the *Total* column, we list the number of tasks that fall within the given year. The 25% column is the same information for the subset that GPT-4 was evaluated on. The remaining model-specific columns contain the % resolved metric.

| # P2P Tests Pass | # F2P Tests Pass | | |
|------------------|------------------|---------|------|
| | All | Partial | None |
| All | Resolved | Partially Resolved | No-Op |
| Partial | Breaking Resolved | Work in Progress | Regression |
| None | Breaking Resolved | Work in Progress | Regression |

Table 22: We present the 6 possible outcomes for a patch generation that is applied successfully and then executed. The outcomes are distinguished by the number of F2P and P2P tests that pass.

successfully. Across all metrics, we find that patch generations across models are much closer in size to the characteristics of average gold edits. While some models still generate fewer lines relative to the corresponding Gold edit (e.g., Claude-2, ChatGPT-3.5, GPT-4), the SWE-Llama models edits are on average longer in most respects.. When considering both Table 8 and Table 24, it becomes clear that models struggle with generating longer output sequences to be correctly formatted patches. Further inspection of such occurrences, as shown in our case studies in Section F, indicate that hallucinations, abiding to existing code style/structure, and referencing long range dependencies correctly are common errors that surface more frequently in longer generations.

## C.7 SOFTWARE ENGINEERING METRICS

We perform preliminary evaluations that explore using *software engineering* metrics to evaluate the efficiency and complexity of large code blocks integrated within a complex codebase. Unlike semantic similarity scoring functions for evaluating fluency and surface form likeness that are popular with traditional NLP benchmarks and have been adopted for code generation, metrics such as Cyclomatic complexity McCabe (1976) and Halstead complexity measures Halstead (1977) are founded upon logical abstractions (e.g., Abstract Syntax Trees) and software principles to quantify the complexity, efficiency, and readability of code as a scalar value. The patch generations and SWE-bench evaluation logs are rich sources of information that software engineering metrics and static analyzers can readily be applied to. Unlike small, code contest benchmarks where the insights of software engineering metrics are not meaningful due to the minuscule scope of the target functionality, SWE-bench's task is complex enough that practitioners can use these tools to gain well-structured, rigorous, and wide-ranging feedback signals on the complexity of a patch generation's change and its effect on the rest of the codebase.

We include our exploratory work here that demonstrates how software engineering metrics can reliably capture characteristics of code quality, and how comparing these statistics across two patches can provide automatic observations about model capabilities. We use the Radon package, a library for computing different software engineering metrics directly from source code.

---

Radon Documentation, open-source codebase

| Model | Claude 2 | ChatGPT-3.5 | GPT-4* | SWE-Llama 7b | SWE-Llama 13b |
|---|---|---|---|---|---|
| Applied | 1078 | 284 | 76 | 1257 | 1196 |
| Resolved | 110 | 12 | 10 | 69 | 91 |
| Breaking Resolved | 26 | 2 | 3 | 17 | 10 |
| Partially Resolved | 15 | 4 | 3 | 17 | 10 |
| Work in Progress | 20 | 2 | 1 | 17 | 16 |
| No-Op | 471 | 174 | 30 | 716 | 672 |
| Regression | 436 | 90 | 29 | 421 | 397 |

Table 23: Categorization of model generations that applied successfully by the cases defined in Table 22. As mentioned, GPT-4 was evaluated on a 25% subset (574 instances) of SWE-bench.

Table 24: Average edits of model generated patches in the oracle retrieval setting across all patches (including unsuccessfully applied patches). For the task instances specific to each model, we calculate the same statistics across the gold patches.

| Model | Total Lines | Added | Removed | Functions | Files |
|---|---|---|---|---|---|
| Claude 2 | 27.2 | 6.6 | 3.3 | 1.2 | 1.1 |
| Gold | 61.6 | 17.8 | 8.6 | 2.6 | 1.4 |
| ChatGPT-3.5 | 42.0 | 6.1 | 3.9 | 1.7 | 1.0 |
| Gold | 44.5 | 12.7 | 5.5 | 2.1 | 1.2 |
| GPT-4 | 22.4 | 4.4 | 1.8 | 0.8 | 0.9 |
| Gold | 50.3 | 14.0 | 6.5 | 2.3 | 1.3 |
| SWE-Llama 13b | 68.9 | 9.5 | 4.3 | 2.5 | 1.6 |
| Gold | 61.5 | 17.8 | 8.6 | 2.6 | 1.4 |
| SWE-Llama 7b | 78.9 | 10.1 | 7.6 | 2.5 | 1.5 |
| Gold | 65.1 | 18.8 | 9.0 | 2.7 | 1.5 |

We look specifically at successfully applied Claude 2 patch predictions in the "Oracle" retrieval setting for the `psf/requests` repository, which several models perform best at as reflected in Figure 4. Per prediction, we apply the patch to the codebase, then calculate the Cyclomatic complexity and Halstead complexity scores for the modified functions. Cyclomatic complexity quantifies the control flow of a function, counting the number of independent execution paths through the source code (McCabe, 1976). A higher Cyclomatic complexity score suggests a more complex function that has higher likelihood of defects and usually suggests difficult maintainability. Halstead complexity counts the number of operators and operands in a program (Halstead, 1977). Per prediction, we also perform the same set of steps for the corresponding gold patch.

We find that software engineering metrics provides automatic qualitative insights into model performance. Consider the following simple case study in Figure 10. While the model patch prediction (left) is fewer lines (6 instead of 11) and modifies fewer files (1 instead of 2) compared to the gold reference solution (right), the model's edit places a conditional within a relatively complex and widely used `HTTPAdapter` class. This introduces two new potential execution outcomes, raising the Cyclomatic complexity of `HTTPAdapter` from 3 to 5. In contrast, while longer, the reference solution imports intra-module dependencies, modifies a logically simpler function in `get_connection`, and defines a new error type `InvalidProxyURL` to capture the novel bug described by the issue.

## D  ADDITIONAL EXPERIMENTAL DETAILS

### D.1  RETRIEVAL DETAILS

**Sparse retrieval.** During retrieval we make a slight augmentation to the documents by pre-pended files' contents with their file paths to better enable retrieval based on filenames that may be mentioned directly in the issue.

> **Problem Statement**: Misleading exception with invalid protocol in proxy variable. When the value of `https_proxy` or `HTTPS_PROXY` variable(s) accidentally miss one '/' in the protocol, a traceback is thrown to the user which doesn't pin point that the issue is with the proxy configuration...

```
                                                   diff --git a/requests/adapters.py b/requests/adapters.py
                                                   --- a/requests/adapters.py
                                                   +++ b/requests/adapters.py
                                                   @@ -300,6 +301,10 @@ def get_connection(self, url, proxies=None):

                                                          if proxy:
                                                              proxy = prepend_scheme_if_needed(proxy, 'http')
                                                   +          proxy_url = parse_url(proxy)
                                                   +          if not proxy_url.host:
                                                   +              raise InvalidProxyURL("Please check proxy URL. It is malformed"
                                                   +                          " and could be missing the host.")
                                                              proxy_manager = self.proxy_manager_for(proxy)
                                                              conn = proxy_manager.connection_from_url(url)
 --- a/requests/adapters.py                            else:
 +++ b/requests/adapters.py                   diff --git a/requests/exceptions.py b/requests/exceptions.py
 @@ -486,6 +486,12 @@ class HTTPAdapter(BaseAdapter):  --- a/requests/exceptions.py
                 low_conn.close()                  +++ b/requests/exceptions.py
                 raise                            @@ -85,6 +85,10 @@ class InvalidHeader(RequestException, ValueError):
                                                       """The header value provided was somehow invalid."""
 +          except (InvalidSchema, MissingSchema) as e:
 +              if 'proxy' in str(e).lower():
 +                  raise ProxyError('Invalid proxy URL: ' + str(e))  +class InvalidProxyURL(InvalidURL):
 +              else:                               +    """The proxy URL provided is invalid."""
 +                  raise                           +
 +                                                  +
           except (ProtocolError, socket.error) as err:   class ChunkedEncodingError(RequestException):
               raise ConnectionError(err, request=request)    """The server declared chunked encoding but sent an invalid chunk."""
```

Figure 10: Comparison of the Claude 2 prediction (left) and reference solution (right) patches for SWE-bench task instance `psf__requests-4356`. While the code generated by the patch is fewer lines of code and solves the problem correctly, the prediction patch introduces greater Cyclomatic complexity (`requests.adapters.py/HTTPAdapter`: $3 \rightarrow 5$) compared to the gold solution (`requests/adapters.py:get_connection`: $2 \rightarrow 3$, `requests/exceptions.py:InvalidHeader`: $0 \rightarrow 1$). Changes that introduce new execution paths are boxed in blue. Parts of the gold patch have been truncated for appearance.

**Oracle retrieval.** Oracle retrieval file paths are simply extracted directly from the reference solution's patch file excluding test files.

## D.2 INFERENCE SETTINGS

Since generations are relatively expensive, we only generate a single patch file per instance. Following precedent in code generation for evaluation in Pass@1 (Chen et al., 2021; Rozière et al., 2023), we simply use greedy decoding for all models.

## D.3 PROMPT TEMPLATE EXAMPLE

Models are prompted with the following general template with slight variations depending on the model used.

```
You will be provided with a partial code base and an issue statement
    explaining a problem to resolve.
<issue>
{ISSUE TEXT}
</issue>


[start of README.md]
{README.md text}
[end of README.md]
[start of file_1.py]
{file_1.py text}
[end of file_1.py]
...


Here is an example of a patch file. It consists of changes to the code
    base. It specifies the file names, the line numbers of each change,
```

and the removed and added lines. A single patch file can contain
changes to multiple files.

```
<patch>
--- a/file.py
+++ b/file.py
@@ -1,27 +1,35 @@
 def euclidean(a, b):
-    while b:
-        a, b = b, a % b
-    return a
+    if b == 0:
+        return a
+    return euclidean(b, a % b)

 def bresenham(x0, y0, x1, y1):
     points = []
     dx = abs(x1 - x0)
     dy = abs(y1 - y0)
-    sx = 1 if x0 < x1 else -1
-    sy = 1 if y0 < y1 else -1
-    err = dx - dy
+    x, y = x0, y0
+    sx = -1 if x0 > x1 else 1
+    sy = -1 if y0 > y1 else 1

-    while True:
-        points.append((x0, y0))
-        if x0 == x1 and y0 == y1:
-            break
-        e2 = 2 * err
-        if e2 > -dy:
+    if dx > dy:
+        err = dx / 2.0
+        while x != x1:
+            points.append((x, y))
             err -= dy
-            x0 += sx
-        if e2 < dx:
-            err += dx
-            y0 += sy
+            if err < 0:
+                y += sy
+                err += dx
+            x += sx
+    else:
+        err = dy / 2.0
+        while y != y1:
+            points.append((x, y))
+            err -= dx
+            if err < 0:
+                x += sx
+                err += dy
+            y += sy

+    points.append((x, y))
     return points
</patch>
```

I need you to solve the provded issue by generating a single patch file
that I can apply directly to this repository using git apply. Please
respond with a single patch file in the format shown above.

Respond below:

Experiments using slightly more or fewer lines of instructions or examples seemed to not affect overall performance substantially, except for the findings of experiments stated in Section 5.

## E    SOCIETAL IMPACT

As reasoning on code has emerged as a foundational skill underlying many LM's capability, a potential future of machine-automated software engineering raises many important questions and has important potential ramifications with regards to AI Safety (Gros et al., 2023). It is important to address questions on how to ensure AI-generated code is faithful to human intents and what guardrails might be in place when human objectives are misinterpreted by code agents that then carry out the task. To observe such problems in a controlled setting and manifest their solutions, we hope SWE-bench might serve as a testbed for designing safe, robust measures towards aligned, verifiable, and safe AI-driven software engineering.

## F    IN-DEPTH ANALYSIS OF SWE-LLAMA GENERATIONS

In this section, we provide five additional qualitative analyses of generations from both Claude 2 and SWE-Llama generations (Oracle retrieval setting) following the style of Section 5.1.

Claude 2 qualitative studies can be found in Tables 25 and 26. Tables 27, 28, and 29 are task instances that Claude 2 did not address correctly. SWE-Llama qualitative studies are covered across Tables 30, 31, 32, 33, 34. For Tables 30, 31, and 32, we present task instances solved correctly by SWE-Llama 13b. In Table 33 and 34, we present two task instances where SWE-Llama 13b does not address the issue correctly, pointing out a subset of the reasoning and generation skills that models may not be adept at enough to accomplish the task at hand.

The observations we make across these sections corroborate with the points stated in the main paper, which is that models tend to struggle with multi-line and multi-file changes, are more adept when the required fix is relatively short, and need help with understanding the codebase in an efficient manner.

Table 25: In this example, Claude 2 correctly addresses an issue from scikit-learn/scikit-learn. However, as demonstrated in the discussion, while the solution is correct, it demonstrates models' tendency to write primitive Python and not employ existing methods within the codebase.

---

**Setting**

- Model: Claude 2 with Oracle retrieval
- Task Instance: `scikit-learn_scikit-learn-13328`
    - Link to Issue: https://github.com/scikit-learn/scikit-learn/pull/13314
    - Link to PR: https://github.com/scikit-learn/scikit-learn/issues/13328

---

**Issue**
TypeError when supplying a boolean X to HuberRegressor fit
#### Description
'TypeError' when fitting 'HuberRegressor' with boolean predictors.

#### Steps/Code to Reproduce

```python
import numpy as np
from sklearn.datasets import make_regression
from sklearn.linear_model import HuberRegressor

# Random data
X, y, coef = make_regression(n_samples=200, n_features=2, noise=4.0, coef
    =True, random_state=0)
X_bool = X > 0
X_bool_as_float = np.asarray(X_bool, dtype=float)

# Works
huber = HuberRegressor().fit(X, y)
# Fails (!)
huber = HuberRegressor().fit(X_bool, y)
# Also works
huber = HuberRegressor().fit(X_bool_as_float, y)
```

#### Expected Results
No error is thrown when 'dtype' of 'X' is 'bool' (second line of code in the snipped above, `.fit(X_bool, y)` ). Boolean array is expected to be converted to 'float' by 'HuberRegressor.fit' as it is done by, say 'LinearRegression'.

#### Actual Results
'TypeError' is thrown:
(Remaining lines omitted)

---

**Gold Patch**

```diff
diff --git a/sklearn/linear_model/huber.py b/sklearn/linear_model/huber.
    py
--- a/sklearn/linear_model/huber.py
+++ b/sklearn/linear_model/huber.py
@@ -251,7 +251,8 @@ def fit(self, X, y, sample_weight=None):
        self : object
        """
        X, y = check_X_y(
-           X, y, copy=False, accept_sparse=['csr'], y_numeric=True)
+           X, y, copy=False, accept_sparse=['csr'], y_numeric=True,
+           dtype=[np.float64, np.float32])
        if sample_weight is not None:
            sample_weight = np.array(sample_weight)
            check_consistent_length(y, sample_weight)
```

**Model Generated Patch**

```
--- a/sklearn/linear_model/huber.py
+++ b/sklearn/linear_model/huber.py
@@ -231,6 +231,9 @@ class HuberRegressor(LinearModel, RegressorMixin,
    BaseEstimator):
        self.tol = tol

+   def _validate_data(self, X, y=None):
+       return X.astype(np.float64) if X.dtype == np.bool else X
+
    def fit(self, X, y, sample_weight=None):
        """Fit the model according to the given training data.

@@ -250,7 +253,7 @@ class HuberRegressor(LinearModel, RegressorMixin,
    BaseEstimator):
        -------
        self : object
        """
-       X, y = check_X_y(
+       X, y = check_X_y(
            X, y, copy=False, accept_sparse=['csr'], y_numeric=True)
        if sample_weight is not None:
            sample_weight = np.array(sample_weight)
@@ -262,6 +265,9 @@ class HuberRegressor(LinearModel, RegressorMixin,
    BaseEstimator):
            raise ValueError(
                "epsilon␣should␣be␣greater␣than␣or␣equal␣to␣1.0,␣got␣%f"
                % self.epsilon)
+
+       X = self._validate_data(X)
```

**Tests**

*Fail to Pass* (1)

✓   sklearn/linear_model/tests/test_huber.py::test_huber_bool

*Pass to Pass* (9)

✓   sklearn/linear_model/tests/test_huber.py::test_huber_max_iter
✓   sklearn/linear_model/tests/test_huber.py::test_huber_gradient
✓   sklearn/linear_model/tests/test_huber.py::test_huber_sample_weights
✓   sklearn/linear_model/tests/test_huber.py::test_huber_sparse
✓   sklearn/linear_model/tests/test_huber.py::test_huber_scaling_invariant

Additional Pass to Pass Tests omitted...

**Discussion.** In this task instance, the issue is describing an issue regarding how types are cast. Specifically, when When fitting the `HuberRegressor` model with boolean predictors, like `X_bool`, a `TypeError` occurs. This issue arises because the `HuberRegressor` does not automatically convert boolean arrays to floats as some other models do, such as `LinearRegression`, causing a type mismatch error. For its fix, the model defines its own `_validate_data` function, which encapsulates the type casting logic, and applies it to `X`. While it passes the tests, this fix could be more efficient and potentially introduces stylistic and organizational inconsistencies. As demonstrated in the reference solution, there is a much simpler solution of passing in `dtype` arguments that leverage the existing flags of the `check_X_y` function call to resolve this more cleanly. Other files' content was not included in the oracle retrieval setting due to the context window limitation.

Table 26: We provide another example where Claude-2 solves the issue correctly and develops a solution similar to the gold patch. However, the gold patch solution is more cognizant of avoiding future potential issues that could be related to this code.

---

**Setting**

- Model: Claude 2 with Oracle retrieval
- Task Instance: `matplotlib_matplotlib-24362`
  - Link to Issue: https://github.com/matplotlib/matplotlib/issues/24349
  - Link to PR: https://github.com/matplotlib/matplotlib/pull/24362

---

**Issue**
[Bug]: sharex and sharey don't accept 0 and 1 as bool values

### Bug summary
When using '0' or '1' in place of 'False' or 'True' in 'sharex' or 'sharex' arguments of 'pyplot.subplots' an error is raised.
### Code for reproduction

```
import matplotlib.pyplot as plt

fig, ax = plt.subplots(ncols=2,sharey=1)
```

### Actual outcome
We get the following error :

```
Traceback (most recent call last):
  (Earlier lines omitted)
  File "/***/matplotlib/lib/matplotlib/gridspec.py", line 293, in subplots
    _api.check_in_list(["all", "row", "col", "none"],
  File "/***/matplotlib/lib/matplotlib/_api/__init__.py", line 131, in
      check_in_list
    raise ValueError(msg)
ValueError: 1 is not a valid value for sharey; supported values are 'all'
    , 'row', 'col', 'none'
```

Using 'sharex' instead of 'sharey' produces the same error (albeit with the following warning:

```
UserWarning: sharex argument to subplots() was an integer. Did you intend
    to use subplot() (without 's')?
```

but this is expected and not part of the present issue)
### Expected outcome
I expected values 1 and 0 to be understood as bool.

---

**Model Generated Patch**

```
diff --git a/lib/matplotlib/gridspec.py b/lib/matplotlib/gridspec.py
index 06dd3f19f6..32ee7c306e 100644
--- a/lib/matplotlib/gridspec.py
+++ b/lib/matplotlib/gridspec.py
@@ -276,9 +276,9 @@ class GridSpecBase:
          raise ValueError("GridSpec.subplots()_only_works_for_GridSpecs_
             "
                     "created_with_a_parent_figure")

-        if isinstance(sharex, bool):
+        if isinstance(sharex, bool) or sharex == 1 or sharex == 0:
            sharex = "all" if sharex else "none"
-        if isinstance(sharey, bool):
+        if isinstance(sharey, bool) or sharey == 1 or sharey == 0:
            sharey = "all" if sharey else "none"
        # This check was added because it is very easy to type
        # 'subplots(1, 2, 1)' when 'subplot(1, 2, 1)' was intended.
```

---

**Gold Patch**

```
diff --git a/lib/matplotlib/gridspec.py b/lib/matplotlib/gridspec.py
--- a/lib/matplotlib/gridspec.py
+++ b/lib/matplotlib/gridspec.py
@@ -276,21 +276,12 @@ def subplots(self, *, sharex=False, sharey=False,
    squeeze=True,
-        if isinstance(sharex, bool):
+        if not isinstance(sharex, str):
            sharex = "all" if sharex else "none"
-        if isinstance(sharey, bool):
+        if not isinstance(sharey, str):
            sharey = "all" if sharey else "none"
-        # This check was added because it is very easy to type
-        # 'subplots(1, 2, 1)' when 'subplot(1, 2, 1)' was intended.
-        # In most cases, no error will ever occur, but mysterious behavior
-        # will result because what was intended to be the subplot index is
-        # instead treated as a bool for sharex. This check should go away
-        # once sharex becomes kwonly.
-        if isinstance(sharex, Integral):
-            _api.warn_external(
-                "sharex argument to subplots() was an integer.  Did you "
-                "intend to use subplot() (without 's')?")
-        _api.check_in_list(["all", "row", "col", "none"],
+
+        _api.check_in_list(["all", "row", "col", "none", False, True],
                           sharex=sharex, sharey=sharey)
```

**Tests**
*Fail to Pass* (1)

✓  `lib/matplotlib/tests/test_subplots.py::test_shared`

*Pass to Pass* (39)

✓  `lib/matplotlib/tests/test_subplots.py::test_label_outer_span`
✓  `lib/matplotlib/tests/test_subplots.py::test_label_outer_non_gridspec`
✓  `lib/matplotlib/tests/test_subplots.py::test_shared_and_moved`
✓  `lib/matplotlib/tests/test_subplots.py::test_exceptions`
✓  `lib/matplotlib/tests/test_subplots.py::test_subplots_offsettext[png]`

Additional Pass to Pass Tests omitted...

**Discussion.** This example asks for 1 and 0 to be treated as boolean values for the `sharex` and `sharey` arguments of the `pyplot.subplots` function. The current code raises an error. In the generated patch, the model adds 1 and 0 as acceptable values. However, the gold patch adds the acceptable values into a list `["all", "row", "col", "none", False, True]`. It also removes the `Integral` related warning tied to this issue. This clean up along with the clarity of the `check_in_list` argument makes the code more concise and readable.

Table 27: In this example, we show an issue from astropy/astropy that Claude 2 does not solve correctly. The error is primarily due to the patch generation's attempt to directly solve the issue.

**Setting**

- Model: Claude 2 with Oracle retrieval
- Task Instance: `astropy_astropy-14365`
    - Link to Issue: https://github.com/astropy/astropy/issues/14363
    - Link to PR: https://github.com/astropy/astropy/pull/14365

**Issue**

ascii.qdp Table format assumes QDP commands are upper case
### Description
ascii.qdp assumes that commands in a QDP file are upper case, for example, for errors they must be "READ SERR 1 2" whereas QDP itself is not case sensitive and case use "read serr 1 2". As many QDP files are created by hand, the expectation that all commands be all-caps should be removed.

### Expected behavior
The following qdp file should read into a 'Table' with errors, rather than crashing.

**read** serr 1 2
1 0.5 1 0.5

### How to Reproduce
Create a QDP file:

```
> cat > test.qdp
read serr 1 2
1 0.5 1 0.5
<EOF>

 > python
Python 3.10.9 (main, Dec 7 2022, 02:03:23) [Clang 13.0.0 (clang
    -1300.0.29.30)] on darwin
Type "help", "copyright", "credits" or "license" for more information.
>>> from astropy.table import Table
>>> Table.read('test.qdp',format='ascii.qdp')
WARNING: table_id not specified. Reading the first available table [
    astropy.io.ascii.qdp]
Traceback (most recent call last):
...
    raise ValueError(f'Unrecognized_QDP_line:_{line}')
ValueError: Unrecognized QDP line: read serr 1 2
```

Running "qdp test.qdp" works just fine.

**Model Generated Patch**

```
--- a/astropy/io/ascii/qdp.py
+++ b/astropy/io/ascii/qdp.py
@@ -60,7 +60,7 @@
    ValueError: Unrecognized QDP line...
    """
    _decimal_re = r"[+-]?(\d+(\.\d*)?|\.\d+)([eE][+-]?\d+)?"
-    _command_re = r"READ [TS]ERR(\s+[0-9]+)+"
+    _command_re = r"read [ts]err(\s+[0-9]+)+"

    sep = delimiter
    if delimiter is None:
```

**Gold Patch**

```
diff --git a/astropy/io/ascii/qdp.py b/astropy/io/ascii/qdp.py
--- a/astropy/io/ascii/qdp.py
+++ b/astropy/io/ascii/qdp.py
@@ -68,7 +68,7 @@ def _line_type(line, delimiter=None):
     _new_re = rf"NO({sep}NO)+"
     _data_re = rf"({_decimal_re}|NO|[-+]?nan)({sep}({_decimal_re}|NO
        |[-+]?nan))*)"
     _type_re = rf"^\s*((?P<command>{_command_re})|(?P<new>{_new_re})|(?P<
        data>{_data_re})?\s*(\!(?P<comment>.*))?\s*$"
-    _line_type_re = re.compile(_type_re)
+    _line_type_re = re.compile(_type_re, re.IGNORECASE)
     line = line.strip()
     if not line:
         return "comment"
@@ -306,7 +306,7 @@ def _get_tables_from_qdp_file(qdp_file,
    input_colnames=None, delimiter=None):

            values = []
            for v in line.split(delimiter):
-                if v == "NO":
+                if v.upper() == "NO":
                    values.append(np.ma.masked)
                else:
                    # Understand if number is int or float
```

**Tests**

*Fail to Pass* (1)

✗  `astropy/io/ascii/tests/test_qdp.py::test_roundtrip[True]`

*Pass to Pass* (8)

✗  `astropy/io/ascii/tests/test_qdp.py::test_get_tables_from_qdp_file`
✗  `astropy/io/ascii/tests/test_qdp.py::test_roundtrip[False]`
✗  `astropy/io/ascii/tests/test_qdp.py::test_read_example`
✗  `astropy/io/ascii/tests/test_qdp.py::test_roundtrip_example`
✗  `astropy/io/ascii/tests/test_qdp.py::test_roundtrip_example_comma`
✓  `astropy/io/ascii/tests/test_qdp.py::test_read_write_simple`
✓  `astropy/io/ascii/tests/test_qdp.py::test_read_write_simple_specify_name`
✓  `astropy/io/ascii/tests/test_qdp.py::test_get_lines_from_qdp`

**Discussion.** This issue requests a fix for handling QDP files; specifically, it asks for the expectation of the commands to be uppercase to be removed. In the model generated patch, the model locates the `READ` and `ERR` commands mentioned in the issue description in a regex pattern and edits it directly. However, within the same file, it is clear that the regex pattern is used in additional patterns, such as `_type_re`, and so a fix to the "root" regex pattern may not be appropriate. In addition, amidst multiple regex pattern variables, identifying the correct one to modify requires reasoning that can potentially be helped by execution. This error in this solution is reflected in the fact that it not only fails to pass the given unit tests, but also fails existing tests enforcing prior behavior.

Table 28: For this issue from the `mwaskom/seaborn` repository, the problem statement includes hyperlinks to images. As discussed in Section 5, a minor subset of SWE-bench tasks include images in them, making image understanding a small but important component to resolving issues that is unexplored by the initial baselines.

---

**Setting**

- Model: Claude 2 with Oracle retrieval
- Task Instance: `mwaskom__seaborn-3217`
    - Link to Issue: https://github.com/mwaskom/seaborn/issues/2907
    - Link to PR: https://github.com/mwaskom/seaborn/pull/3217

---

**Issue**

Width computation after histogram slightly wrong with log scale Note the slight overlap here:

```
(
    so.Plot(tips, "total_bill")
    .add(so.Bars(alpha=.3, edgewidth=0), so.Hist(bins=4))
    .scale(x="log")
)
```

It becomes nearly imperceptible with more bins:

```
(
    so.Plot(tips, "total_bill")
    .add(so.Bars(alpha=.3, edgewidth=0), so.Hist(bins=8))
    .scale(x="log")
)
```

This is not about 'Bars'; 'Bar' has it too:

```
(
    so.Plot(tips, "total_bill")
    .add(so.Bar(alpha=.3, edgewidth=0, width=1), so.Hist(bins=4))
    .scale(x="log")
)
```

---

**Model Generated Patch**

```
--- a/seaborn/_core/plot.py
+++ b/seaborn/_core/plot.py
@@ -1327,9 +1327,7 @@
                elif "width" in df:
                    view_width = view_df["width"]
                else:
-                   view_width = 0.8 # TODO what default?
-               spacing = scales[orient]._spacing(view_df.loc[view_idx,
    orient])
-               width.loc[view_idx] = view_width * spacing
+                   width.loc[view_idx] = view_width
            df["width"] = width

        if "baseline" in mark._mappable_props:
```

---

**Gold Patch**

```
diff --git a/seaborn/_core/plot.py b/seaborn/_core/plot.py
--- a/seaborn/_core/plot.py
+++ b/seaborn/_core/plot.py
@@ -1377,10 +1377,9 @@ def _unscale_coords(
-        drop_cols = [*coord_cols, "width"] if "width" in df else
   coord_cols
         out_df = (
             df
-            .drop(drop_cols, axis=1)
+            .drop(coord_cols, axis=1)
             .reindex(df.columns, axis=1) # So unscaled columns retain their
                 place
@@ -1396,12 +1395,6 @@ def _unscale_coords(
-            if var == orient and "width" in view_df:
-                width = view_df["width"]
-                out_df.loc[values.index, "width"] = (
-                    transform(values + width / 2) - transform(values -
   width / 2)
-                )
-
        return out_df
diff --git a/seaborn/_marks/bar.py b/seaborn/_marks/bar.py
--- a/seaborn/_marks/bar.py
+++ b/seaborn/_marks/bar.py
@@ -29,17 +29,23 @@ class BarBase(Mark):
+        transform = scales[orient]._matplotlib_scale.get_transform()
+        forward = transform.transform
+        reverse = transform.inverted().transform
+
+        other = {"x": "y", "y": "x"}[orient]
+
+        pos = reverse(forward(data[orient]) - data["width"] / 2)
+        width = reverse(forward(data[orient]) + data["width"] / 2) - pos
+
+        val = (data[other] - data["baseline"]).to_numpy()
+        base = data["baseline"].to_numpy()
+
(Remaining lines omitted...)
```

**Tests**

*Fail to Pass* (1)

✗  tests/_marks/test_bar.py::TestBars::test_log_scale

*Pass to Pass* (8)

✗  tests/_marks/test_bar.py::TestBar::test_categorical_positions_vertical
✗  tests/_marks/test_bar.py::TestBar::test_categorical_positions_horizontal
✗  tests/_marks/test_bar.py::TestBar::test_numeric_positions_vertical
✗  tests/_marks/test_bar.py::TestBar::test_numeric_positions_horizontal
✗  tests/_marks/test_bar.py::TestBar::test_set_properties

Additional Pass to Pass Tests omitted...

**Discussion.** The baseline model's inability to solve this issue can be directly attributed to a lack of an image input. We include this example to demonstrate that debugging real software issues may involve digesting images; this presents a whole separate body of understanding and reasoning-related challenges. Beyond the SWE-bench task, additional problems in software engineering, such as interpreting system design diagram, user interface specifications, and user feedback presents interesting vision-language challenges.

Table 29: In this final example of a Claude 2 generation, the model must resolve an error related to resolving an error regarding cyclic dependencies. Claude 2's solution under-delivers on an otherwise complex problem.

**Setting**

- Model: Claude 2 with Oracle retrieval
- Task Instance: sympy__sympy-18211
    - Link to Issue: https://github.com/pylint-dev/pylint/issues/3525
    - Link to PR: https://github.com/pylint-dev/pylint/pull/4703

**Issue**
Imports within TYPE_CHECKING can induce "cyclic-import"
Suppose you have two modules: a.py and b.py. 'a' imports 'b', but 'b' needs 'a' in order to do type annotations, then this is usually done by importing 'a' within a 'TYPE_CHECKING' block. Unfortunately, this causes pylint to report 'cyclic-import'.

Possibly related to https://github.com/PyCQA/pylint/issues/3285

**Gold Patch**

```
diff --git a/pylint/checkers/imports.py b/pylint/checkers/imports.py
--- a/pylint/checkers/imports.py
+++ b/pylint/checkers/imports.py
@@ -813,7 +813,9 @@ def _get_imported_module(self, importnode, modname):
-    def _add_imported_module(self, node, importedmodname):
+    def _add_imported_module(
+        self, node: Union[astroid.Import, astroid.ImportFrom],
    importedmodname: str
+    ) -> None:
        """notify an imported module, used to analyze dependencies"""
        module_file = node.root().file
        context_name = node.root().name
@@ -826,6 +828,10 @@ def _add_imported_module(self, node, importedmodname
    ):
        except ImportError:
            pass

+        in_type_checking_block = (
+            isinstance(node.parent, astroid.If) and node.parent.
    is_typing_guard()
+        )
+
        if context_name == importedmodname:
            self.add_message("import-self", node=node)

@@ -845,7 +851,10 @@ def _add_imported_module(self, node, importedmodname
    ):

        # update import graph
        self.import_graph[context_name].add(importedmodname)
-        if not self.linter.is_message_enabled("cyclic-import", line=
    node.lineno):
+        if (
+            not self.linter.is_message_enabled("cyclic-import", line=
    node.lineno)
+            or in_type_checking_block
+        ):
            self._excluded_edges[context_name].add(importedmodname)
```

**Model Generated Patch**

```
diff --git a/pylint/checkers/imports.py b/pylint/checkers/imports.py
index 4f3eed2..2b74a8d 100644
--- a/pylint/checkers/imports.py
+++ b/pylint/checkers/imports.py
@@ -49,6 +49,7 @@ import copy
 import os
 import sys
 from distutils import sysconfig
+from typing import TYPE_CHECKING
 from typing import Dict, List, Union

 import astroid
@@ -61,6 +62,9 @@ from pylint.checkers.utils import (
     node_ignores_exception,
 )
 from pylint.exceptions import EmptyReportError
+if TYPE_CHECKING:
+    from pylint.lint import PyLinter
+
 from pylint.graph import DotBackend, get_cycles
 from pylint.interfaces import IAstroidChecker
 from pylint.reporters.ureports.nodes import Paragraph, VerbatimText,
     VNode
```

**Tests**

*Fail to Pass* (1)

✗  `tests/_marks/test_bar.py::TestBars::test_log_scale`

*Pass to Pass* (11)

✓  `tests/test_func.py::test_functionality[func_w0401_disabled_in_func.py]`
✓  `tests/test_func.py::test_functionality[func_i0022.py]`
✓  `tests/test_func.py::test_functionality[func_w0401.py]`
✓  `tests/test_func.py::test_functionality[func_w0801.py]`
✓  `tests/test_func.py::test_functionality[func_w0401_package]`

Additional Pass to Pass Tests omitted...

**Discussion.** In this problem, the model is being asked to provide `TYPE_CHECKING` based handling for resolving cyclic imports, which pylint currently throws an error for. The gold patch has a detailed solution that involves conditional handling within the `TYPE_CHECKING` block, excludes the cyclic import messages, and has generally more specific handling. The model generated patch employs a "Stack Overflow" like solution; while creating a `TYPE_CHECKING` block is a popular solution for resolving imports, solving this issue for the codebase in particular requires a more nuanced solution than a general answer that may be available online.

Table 30: Example of a SWE-Llama 13b generation that correctly solves a SWE-bench task instance. In this example, the SWE-Llama generation is exactly the same as the solution.

---

**Setting**

- Model: SWE-Llama 13B with Oracle retrieval
- Task Instance: pydata_xarray-5131
    - Link to Issue: https://github.com/pydata/xarray/pull/5130
    - Link to PR: https://github.com/pydata/xarray/pull/5131

---

**Issue**

When displaying a DatasetGroupBy in an interactive Python session, the first line of output contains a trailing whitespace. The first example in the documentation demonstrate this:

```
>>> import xarray as xr, numpy as np
>>> ds = xr.Dataset(
...    {"foo": (("x", "y"), np.random.rand(4, 3))},
...    coords={"x": [10, 20, 30, 40], "letters": ("x", list("abba"))},
... )
>>> ds.groupby("letters")
DatasetGroupBy, grouped over 'letters'
2 groups with labels 'a', 'b'.
```

There is a trailing whitespace in the first line of output which is "DatasetGroupBy, grouped over 'letters' ". This can be seen more clearly by converting the object to a string (note the whitespace before ⌴n⌴ ):

```
>>> str(ds.groupby("letters"))
"DatasetGroupBy,⌴grouped⌴over⌴'letters'⌴\n2⌴groups⌴with⌴labels⌴'a',⌴'b'."
```

While this isn't a problem in itself, it causes an issue for us because we use flake8 in continuous integration to verify that our code is correctly formatted... [20 more lines]

---

**Gold Patch**

```
diff --git a/xarray/core/groupby.py b/xarray/core/groupby.py
--- a/xarray/core/groupby.py
+++ b/xarray/core/groupby.py
@@ -436,7 +436,7 @@ def __iter__(self):
        return zip(self._unique_coord.values, self._iter_grouped())

    def __repr__(self):
-       return "{},⌴grouped⌴over⌴{!r}⌴\n{!r}⌴groups⌴with⌴labels⌴{}.".format(
+       return "{},⌴grouped⌴over⌴{!r}\n{!r}⌴groups⌴with⌴labels⌴{}.".format(
            self.__class__.__name__,
```

---

**Model Generated Patch**

```
diff --git a/xarray/core/groupby.py b/xarray/core/groupby.py
--- a/xarray/core/groupby.py
+++ b/xarray/core/groupby.py
@@ -436,7 +436,7 @@ def __iter__(self):
        return zip(self._unique_coord.values, self._iter_grouped())

    def __repr__(self):
-       return "{},⌴grouped⌴over⌴{!r}⌴\n{!r}⌴groups⌴with⌴labels⌴{}.".format(
+       return "{},⌴grouped⌴over⌴{!r}\n{!r}⌴groups⌴with⌴labels⌴{}.".format(
            self.__class__.__name__,
```

---

**Tests**
*Fail to Pass* (10)

✓    `xarray/tests/test_groupby.py::test_groupby_repr[obj0-x]`
✓    `xarray/tests/test_groupby.py::test_groupby_repr[obj0-y]`
✓    `xarray/tests/test_groupby.py::test_groupby_repr[obj0-z]`
✓    `xarray/tests/test_groupby.py::test_groupby_repr[obj0-month]`
✓    `xarray/tests/test_groupby.py::test_groupby_repr[obj1-x]`
✓    `xarray/tests/test_groupby.py::test_groupby_repr[obj1-y]`
✓    `xarray/tests/test_groupby.py::test_groupby_repr[obj1-z]`
✓    `xarray/tests/test_groupby.py::test_groupby_repr[obj1-month]`
✓    `xarray/tests/test_groupby.py::test_groupby_repr_datetime[obj0]`
✓    `xarray/tests/test_groupby.py::test_groupby_repr_datetime[obj1]`

*Pass to Pass* (16)

✓    `xarray/tests/test_groupby.py::test_consolidate_slices`
✓    `xarray/tests/test_groupby.py::test_groupby_da_datetime`
✓    `xarray/tests/test_groupby.py::test_groupby_duplicate_coordinate_labels`
✓    `xarray/tests/test_groupby.py::test_groupby_input_mutation`
✓    `xarray/tests/test_groupby.py::test_groupby_map_shrink_groups[obj0]`

Additional Pass to Pass Tests omitted...

**Discussion.** In this problem, the model is asked to correct a trailing whitespace issue, and the issue shows a demonstration of the error that arises during usage, but does not explicitly point out the function to fix in the model. In the oracle retrieval setting, the model is given the `xarray/core/groupby.py` file to edit directly, but most locate the line to edit. In this situation, the model successfully reasons that in the interactive Python setting, the underlying function that is being invoked is actually the `__repr__` function. After locating the function, it must figure out where in the `return` statement the whitespace is located. At this point, the fix is quite evident, as there is an explicit space that the model removes. In this case, the model successfully passes all tests for verifying the correctness of this fix explicitly (10 *fail* to *pass* tests) while also maintaining the existing behavior of the file (16 *pass* to *pass* tests).

Table 31: Example of a SWE-Llama 13b generation that correctly solves a SWE-bench task instance. In this example, the SWE-Llama generation is different from the gold patch.

**Setting**

- Model: SWE-Llama 13B with Oracle retrieval
- Task Instance: `psf_requests-1724`
    - Link to Issue: https://github.com/psf/requests/issues/1723
    - Link to PR: https://github.com/psf/requests/pull/1724

**Issue**

Unicode method names cause UnicodeDecodeError for some requests in Python 2.7.2 The following example works fine:

```
files = {u'file': open(u'/usr/bin/diff', u'rb')}
response = requests.request(method='POST', url=u'http://httpbin.org/post'
    ,
    files=files)
```

But the following example (using 'method=u'POST'' instead of 'method='POST'') produces a UnicodeDecodeError:

```
files = {u'file': open(u'/usr/bin/diff', u'rb')}
response = requests.request(method=u'POST', url=u'http://httpbin.org/post
    ',
    files=files)

Traceback (most recent call last):
  [Full stack trace hidden]
  File "/System/Library/Frameworks/Python.framework/Versions/2.7/...
    msg += message_body
UnicodeDecodeError: 'ascii' codec can't decode byte 0xcf in position 140:
    ordinal not in range(128)
```

My guess is that 'u'POST'' is infecting the header with unicode when it should be a string. This is because 'sessions.py:313' is simply:

```
req.method = method.upper()
```

[Additional lines hidden...]

**Gold Patch**

```
diff --git a/requests/sessions.py b/requests/sessions.py
--- a/requests/sessions.py
+++ b/requests/sessions.py
@@ -12,7 +12,7 @@
 from collections import Mapping
 from datetime import datetime
-from .compat import cookielib, OrderedDict, [3 more omitted]
+from .compat import cookielib, OrderedDict, [3 omitted,
+  builtin_str
 from .cookies import cookiejar_from_dict, extract_cookies_to_jar,
   RequestsCookieJar
 from .models import Request, PreparedRequest
 from .hooks import default_hooks, dispatch_hook
@@ -309,6 +309,9 @@ def request(self, method, url,
       :param cert: (optional) if String, path to ssl client cert
         file (.pem). If Tuple, ('cert', 'key') pair.
       """
+
+      method = builtin_str(method)
```

**Model Generated Patch**

```
diff --git a/requests/sessions.py b/requests/sessions.py
--- a/requests/sessions.py
+++ b/requests/sessions.py
@@ -254,7 +254,7 @@ def prepare_request(self, request):

        p = PreparedRequest()
        p.prepare(
-           method=request.method.upper(),
+           method=str(request.method).upper(),
            url=request.url,
            files=request.files,
            data=request.data,
```

**Tests**

*Fail to Pass* (6)

- ✓ `test_requests.py::RequestsTestCase::test_DIGEST_AUTH_RETURNS_COOKIE`
- ✓ `test_requests.py::RequestsTestCase::test_DIGEST_HTTP_200_OK_GET`
- ✓ `test_requests.py::RequestsTestCase::test_different_encodings_dont_break`
- ✓ `test_requests.py::RequestsTestCase::test_generic_cookiejar_works`
- ✓ `test_requests.py::RequestsTestCase::test_uppercase_scheme_redirect`

*Pass to Pass* (75)

- ✓ `test_requests.py::RequestsTestCase::test_DIGEST_AUTH_SETS_SESSION_COOKIES`
- ✓ `test_requests.py::RequestsTestCase::test_DIGEST_STREAM`
- ✓ `test_requests.py::RequestsTestCase::test_HTTP_200_OK_GET_ALTERNATIVE`
- ✓ `test_requests.py::RequestsTestCase::test_HTTP_200_OK_GET_WITH_PARAMS`
- ✓ `test_requests.py::RequestsTestCase::test_HTTP_200_OK_HEAD`

Additional Pass to Pass Tests omitted...

**Discussion.** Unlike the problem presented in Table 30, instead of an interactive python input, the model is given an error trace from a function invocation, where the steps to recreate the problem from a user's standpoint is offered. In addition, the issue also provides a suggestion during the last line on a potential fix for the problem. Once again in the oracle setting, the model takes advantage of the natural language suggestion and localizes the change to a line that is semantically quite similar to the suggestion ( `req.method = method.upper()` compared to `method=str(request.method).upper()` . The model correctly addresses the issue, as reflected by all the passing tests. However, in this situation, we can see that the model is actually quite "lucky" in its success. First, while the gold patch leverages built in functions that are used for uniformity in handling string formatting, the model's decision to use more primitive Python to tackle this issue would likely not be well-received by a human reviewer due to its inconsistency with the rest of the codebase. Further inspection of the `builtin_str` function, not shown in this figure, also reveals that it handles additional string formatting cases beyond what was portrayed in the issue. While the model solves the task this time, it is questionable whether this solution is robust to future issues.

Table 32: This is another example where SWE-Llama13b solves the task successfully. This example is interesting because the model develops a somewhat novel solution compared to the reference that is arguably more efficient and cleaner.

**Setting**

- Model: SWE-Llama 13B with Oracle retrieval
- Task Instance: `django__django-11727`
    - Link to Issue: https://code.djangoproject.com/ticket/29376
    - Link to PR: https://github.com/django/django/pull/11727

**Issue**

Allow hiding the "Save and Add Another" button with a `show_save_and_add_another` context variable

To provide better adjustability, to introduce new context var - `show_save_and_add_another`. E.g. if I want to hide button "Save and add another", I can just modify `extra_context` - write `False` to the variable. For other buttons - "Save" and "Save and continue editing", this already works exactly in this manner.

**Gold Patch**

```
diff --git a/django/contrib/admin/templatetags/admin_modify.py b/django
    /...
--- a/django/contrib/admin/templatetags/admin_modify.py
+++ b/django/contrib/admin/templatetags/admin_modify.py
@@ -54,12 +54,20 @@ def submit_row(context):
    is_popup = context['is_popup']
    save_as = context['save_as']
    show_save = context.get('show_save', True)
+   show_save_and_add_another =
        context.get('show_save_and_add_another', True)
(7 lines omitted...)
        or (has_add_permission and add)
        or has_editable_inline_admin_formsets
+   can_save_and_add_another = (
+       has_add_permission and
+       not is_popup and
+       (not save_as or add) and
+       can_save and
+       show_save_and_add_another
+   )
    can_save_and_continue = not is_popup and can_save and
        has_view_permission
        and show_save_and_continue
    can_change = has_change_permission or
        has_editable_inline_admin_formsets
    ctx = Context(context)
@@ -70,10 +78,7 @@ def submit_row(context):
        change and context.get('show_delete', True)
    ),
    'show_save_as_new': not is_popup and has_change_permission
        and change and save_as,
-   'show_save_and_add_another': (
-       has_add_permission and not is_popup and
-       (not save_as or add) and can_save
-   ),
+   'show_save_and_add_another': can_save_and_add_another,
    'show_save_and_continue': can_save_and_continue,
    'show_save': show_save and can_save,
```

**Model Generated Patch**

```
diff --git a/django/contrib/admin/templatetags/admin_modify.py b/django
    /...
--- a/django/contrib/admin/templatetags/admin_modify.py
+++ b/django/contrib/admin/templatetags/admin_modify.py
@@ -72,7 +72,7 @@ def submit_row(context):
        'show_save_as_new': not is_popup and has_change_permission
            and change and save_as,
        'show_save_and_add_another': (
            has_add_permission and not is_popup and
-           (not save_as or add) and can_save
+           (not save_as or add) and can_save and context
                .get('show_save_and_add_another', True)
        ),
        'show_save_and_continue': can_save_and_continue,
        'show_save': show_save and can_save,
```

**Tests**

*Fail to Pass* (2)

✓   `test_override_show_save_and_add_another`
✓   `test_submit_row (admin_views.test_templatetags.Admin...)`

*Pass to Pass* (3)

✓   `test_choice_links (admin_views.test_templatetags.DateHierarchyTests)`
✓   `test_override_change_form_template_tags`
✓   `test_override_change_list_template_tags`

**Discussion.** For this task instance that comes from the django/django repository, the model is asked to introduce a context variable that would allow a user to hide the "Save and Add Another" button via a context variable, similar to how it is done for two other existing buttons. The task is a bit more difficult compared to the prior two settings because no explicit stack trace or programmatic demonstration of the issue is offered. In this relatively under-specified setting that does not provide suggestion with regards to localizing the function correctly, the model successfully reasons that it should adjust the existing `show_save_and_add_another` key/value pair. When comparing the gold patch solution, it can be argued that the model generated patch produces a much more efficient solution in terms of lines edited, as it makes the smallest edit necessary to incorporate `context` as a flag for setting the `show_save_and_add_another` hidden status. However, similar to the discussion in Table 31, stylistically, the gold patch edits are much more consistent with the codebase, and additional changes that are not explicitly discussed in the issue are also made to adhere to what has been done in the codebase (i.e. the addition of a `can_save_and_add_another field` . This task is an example of a potentially exciting direction where via human guidance or better understanding of a codebase in general, models would then adjust their generations to not just make the *functionally* correct changes, but also the *stylistically* right ones, too.

Table 33: This is an example where SWE-Llama13b writes an incorrect solution, but maintains prior behavior successfully. The difference in the model's generation and the gold patch here demonstrates how models tend to under-generate the necessary fixes.

---

**Setting**

- Model: SWE-Llama 13B with Oracle retrieval
- Task Instance: `sphinx-doc__sphinx-11445`
    - Link to Issue: https://github.com/sphinx-doc/sphinx/issues/11437
    - Link to PR: https://github.com/sphinx-doc/sphinx/pull/11445

---

**Issue**

Using `rst_prolog` removes top level headings containing a domain directive

### Describe the bug

If `rst_prolog` is set, then any documents that contain a domain directive as the first heading (eg `:mod:`) do not render the heading correctly or include the heading in the toctree. In the example below, if the heading of `docs/mypackage.rst` were `mypackage2` instead of `:mod:mypackage2` then the heading displays correctly. Similarly, if you do not set `rst_prolog` then the heading will display correctly. This appears to have been broken for some time because I can reproduce it in v4.0.0 of Sphinx

### How to Reproduce

```
$ sphinx-quickstart --no-sep --project mypackage --author me -v 0.1.0
      --release 0.1.0 --language en docs
$ echo -e 'Welcome\n=======\n\n..␣toctree::\n\n␣␣␣mypackage\n'
      > docs/index.rst
$ echo -e ':mod:`mypackage2`\n=================\n\nContent\n\n
␣␣␣␣␣␣␣␣Subheading\n----------\n' > docs/mypackage.rst
$ echo -e 'rst_prolog␣=␣"""\n..␣|psf|␣replace::␣Python␣Software
␣␣␣␣␣␣␣␣Foundation\n"""\n' >> docs/conf.py
$ sphinx-build -b html . _build
$ grep 'mypackage2' docs/_build/index.html
```

---

**Model Generated Patch**

```
diff --git a/sphinx/util/rst.py b/sphinx/util/rst.py
--- a/sphinx/util/rst.py
+++ b/sphinx/util/rst.py
@@ -25,7 +25,7 @@

 logger = logging.getLogger(__name__)
-docinfo_re = re.compile(':\\w+:.*?')
+docinfo_re = re.compile(':\\w+:.*?\n')
 symbols_re = re.compile(r'([!-\-/:-@\[-`{-~])')
 SECTIONING_CHARS = ['=', '-', '~']
```

---

**Tests**

*Fail to Pass* (2)

✗ `tests/test_util_rst.py::test_prepend_prolog_with_roles_with_newline`
✗ `tests/test_util_rst.py::test_prepend_prolog_with_roles_without_newline`

*Pass to Pass* (8)

✓ `tests/test_util_rst.py::test_prepend_prolog`
✓ `tests/test_util_rst.py::test_prepend_prolog_with_CR`
✓ `tests/test_util_rst.py::test_prepend_prolog_without_CR`

Additional Pass to Pass Tests omitted...

---

**Gold Patch**

```
diff --git a/sphinx/util/rst.py b/sphinx/util/rst.py
--- a/sphinx/util/rst.py
+++ b/sphinx/util/rst.py
@@ -10,22 +10,17 @@

 from docutils.parsers.rst import roles
 from docutils.parsers.rst.languages import en as english
+from docutils.parsers.rst.states import Body
 from docutils.statemachine import StringList
 from docutils.utils import Reporter
-from jinja2 import Environment
+from jinja2 import Environment, pass_environment

 from sphinx.locale import __
 from sphinx.util import docutils, logging

-try:
-    from jinja2.utils import pass_environment
-except ImportError:
-    from jinja2 import environmentfilter as pass_environment
-
-
 logger = logging.getLogger(__name__)

-docinfo_re = re.compile(':\\w+:.*?')
+FIELD_NAME_RE = re.compile(Body.patterns['field_marker'])
 symbols_re = re.compile(r'([!-\-/:-@\[-`{-~])') # symbols without dot(0
     x2e)
 SECTIONING_CHARS = ['=', '-', '~']

@@ -80,7 +75,7 @@ def prepend_prolog(content: StringList, prolog: str) ->
     None:
     if prolog:
         pos = 0
         for line in content:
-            if docinfo_re.match(line):
+            if FIELD_NAME_RE.match(line):
                 pos += 1
             else:
                 break
@@ -91,6 +86,7 @@ def prepend_prolog(content: StringList, prolog: str) ->
     None:
             pos += 1

         # insert prolog (after docinfo if exists)
+        lineno = 0
         for lineno, line in enumerate(prolog.splitlines()):
             content.insert(pos + lineno, line, '<rst_prolog>', lineno)
```

**Discussion.** For this task instance from the `sphinx-doc/sphinx` repository, a model is asked to write logic to fix a case where the title is incorrectly being rendered. Simply understanding the jargon being used and mapping such words to logic within the codebase is a significant challenge faced by the model. The model is given a command line call that can help with this, but grounding the terminology presented in the issues within the codebase is essential. From comparing the gold patch and model generated patch, it is clear that the model does not come close to solving the task. The model does generally identify that fixing the regex pattern is the correct action, as this is what the gold patch does, too. However, where the model and oracle retrieval setting collectively fall short is mainly due to the significant use of additional modules from both the codebase itself and third party libraries. This example highlights the importance and potential for training language models and designing inference procedures that allow for the automated discovery of such information.

Table 34: In this final example, SWE-Llama 13b not only does not solve the task, but also corrupts existing behavior in the model. This example demonstrates the need for models to understand the codebase beyond the scope of the required edits.

---

**Setting**

- Model: SWE-Llama 13B with Oracle retrieval
- Task Instance: `scikit-learn__scikit-learn-13241`
    - Link to Issue: https://github.com/scikit-learn/scikit-learn/issues/8798
    - Link to PR: https://github.com/scikit-learn/scikit-learn/pull/13241

---

**Issue**
Differences among the results of KernelPCA with rbf kernel

#### Description
When I run KernelPCA for dimension reduction for the same datasets, the results are different in signs.

#### Steps/Code to Reproduce
Just to reduce the dimension to 7 with rbf kernel:
```
pca = KernelPCA(n_components=7, kernel='rbf', copy_X=False, n_jobs=-1)
pca.fit_transform(X)
```

#### Expected Results
The same result.

#### Actual Results
The results are the same except for their signs:
```
[[-0.44457 -0.18155 -0.10873 0.13548 -0.14371 -0.05746 0.18124]]
[[ 0.44457 0.18155 0.10873 -0.13548 -0.14371 -0.05746 -0.18124]]
[[-0.44457 -0.18155 0.10873 0.13548 0.14371 0.057469 0.18124]]
```

---

**Gold Patch**
```
diff --git a/sklearn/decomposition/kernel_pca.py b/sklearn/decompos...
--- a/sklearn/decomposition/kernel_pca.py
+++ b/sklearn/decomposition/kernel_pca.py
@@ -8,6 +8,7 @@
 from scipy.sparse.linalg import eigsh

 from ..utils import check_random_state
+from ..utils.extmath import svd_flip
 from ..utils.validation import check_is_fitted, check_array
 from ..exceptions import NotFittedError
 from ..base import BaseEstimator, TransformerMixin,
     _UnstableOn32BitMixin
@@ -210,6 +211,10 @@ def _fit_transform(self, K):
                                      maxiter=self.max_iter,
                                      v0=v0)

+        # flip eigenvectors' sign to enforce deterministic output
+        self.alphas_, _ = svd_flip(self.alphas_,
+                          np.empty_like(self.alphas_).T)
+
         # sort eigenvectors in descending order
         indices = self.lambdas_.argsort()[::-1]
         self.lambdas_ = self.lambdas_[indices]
```

**Model Generated Patch**

```
diff --git a/sklearn/decomposition/kernel_pca.py b/sklearn/decompos...
--- a/sklearn/decomposition/kernel_pca.py
+++ b/sklearn/decomposition/kernel_pca.py
@@ -220,7 +220,7 @@ def _fit_transform(self, K):
            self.alphas_ = self.alphas_[:, self.lambdas_ > 0]
            self.lambdas_ = self.lambdas_[self.lambdas_ > 0]

-       return K
+       return K / self.lambdas_

    def _fit_inverse_transform(self, X_transformed, X):
        if hasattr(X, "tocsr"):
```

**Tests**

*Fail to Pass* (1)

✗  `sklearn/tests/test_kernel_pca.py::test_kernel_pca_deterministic_output`

*Pass to Pass* (54)

✗  `sklearn/tests/test_kernel_pca.py::test_kernel_pca`
✗  `sklearn/tests/test_kernel_pca.py::test_kernel_pca_consistent_transform`
✗  `sklearn/tests/test_kernel_pca.py::test_kernel_pca_deterministic_output`
✗  `sklearn/tests/test_kernel_pca.py::test_kernel_pca_sparse`
✗  `sklearn/tests/test_kernel_pca.py::test_kernel_pca_linear_kernel`
✗  `sklearn/tests/test_kernel_pca.py::test_kernel_pca_n_components`
✗  `sklearn/tests/test_kernel_pca.py::test_remove_zero_eig`
✗  `sklearn/tests/test_kernel_pca.py::test_kernel_pca_precomputed`
✗  `sklearn/tests/test_kernel_pca.py::test_gridsearch_pipeline`
✗  `sklearn/tests/test_kernel_pca.py::test_gridsearch_pipeline_precomputed`
✓  `sklearn/tests/test_pca.py::test_pca_dtype_preservation[randomized]`
✓  `sklearn/tests/test_pca.py::test_pca_dtype_preservation[auto]`
✓  `sklearn/tests/test_pca.py::test_pca_deterministic_output`

Additional Pass to Pass Tests omitted...

**Discussion.** In this example from the `scikit-learn/scikit-learn` library, a model is given an observation that the parities of the output from a call to the `fit_transform` function are flipped. The model is also given lines of Python code and its standard output that recreates the issue. The gold patch imports and uses the `svd_flip` function to solve this issue within a different line of the `_fit_transform` function. What's different about the model's failure for this task beyond the points discussed for the Table 33 example is that, in addition to understanding third party dependencies that its edits rely on, it is also important for a model to understand what other parts of the codebase in turn depend on the function it is changing. This example presents a different facet as to why processing long contexts extend beyond the local edit scope is a difficult but worthwhile challenge.

