# OpenReview forum: "SWE-bench: Can Language Models Resolve Real-world Github Issues?"
_ICLR.cc/2024/Conference — ICLR 2024 oral_

### Official Review · Reviewer_BfZn · 2023-10-29

**Soundness:** 2 fair
**Presentation:** 3 good
**Contribution:** 2 fair
**Rating:** 5
**Confidence:** 3

**Summary:**

This paper introduces a new benchmark, SWE-bench, which collects code and issues from 12 Python repositories. This benchmark also considers the convenience of subsequent evaluation, and the test code for relevant issues is included. Moreover, this paper also finetunes Code Llama with SWE-bench training data. Experimental results show that there are still many challenges for existing LLM to solve real-world issues.

**Strengths:**

1.	The paper is generally well-written.
2.	This paper introduced a new dataset SWE-bench that contains 2294 GitHub issues and related test scripts. The dataset can be used to evaluate the methods for resolving real-world GitHub issues.

**Weaknesses:**

1.	Some of the comparison is not very fair. As Claude 2 is trained on data up to early 2023, GPT's knowledge cutoff is September 2021 and there is no specific time for Code Llama’s training data, evaluating these models on the dataset that contains instances before 2023 is not fair enough.
2.	The contribution of SWE-Llama is not significant, especially for an AI conference. The paper could better target a software engineering/programming conference.
3.	This method is mainly based on Code Llama while there is no comparison between Code Llama and SWE-Llama.
4.	Some of the experimental analysis is not solid enough. For example, in the “Difficulty correlates with output length” (Section 5), Table 8 only presents all successfully applied patches, and does not show the correlation between difficulty and output length. The length of other patches needs to be taken into account.
5.	There are a lot of work on automated bug fixing, including LLM-based ones and traditional ones. The authors could discuss and compare. For example:
Jiang et al., Shaping Program Repair Space with Existing Patches and Similar Code, Proc. ISSTA 2018.
D. Sobania, et al., An analysis of the automatic bug fixing performance of Chatgpt,arXiv:2301.08653, 2023.

**Questions:**

1.	As the experimental results of GPT-4 are on a 20% random subset of SWE-bench while there is no comparison of other models on the same subset. If we only look at this part of the subset, are all the conclusions in the paper still valid/consistent?
2.	Why are these 12 Python repositories chosen as the source of the benchmark? Does the selection of the programming language and repository influence the results of the comparison?

---

> ### Author Response · Authors · 2023-11-15
>
> Thanks for your thorough review of the paper and suggestions - the feedback has been helpful to clarify several details.
>
> Please refer to the general response for a summary of our updates, which are also reflected in the updated paper. We also answer your specific concerns as follows:
>
> 1. **Temporal comparisons to determine if knowledge cutoff provides unfair advantage**: We agree that comparing different models trained on data from different points in time could lead to unfair advantages, which is a shared concern among all LLM research.
> To fully investigate this, we added an extended temporal analysis on our results (Appendix C.3). We calculate model performance on task for each calendar year from 2018-2023.
>
> | Year | Total | 25\% | Claude 2 | GPT-3.5 | GPT-4 | SWE-Llama 13b | SWE-Llama 7b |
> |-|-|-|-|-|-|-|-|
> | 2023    | 244 | 61  | 4.51 | 1.56 | 0.00 | 4.07 | 3.50 |
> | 2022    | 395 | 117 | 4.05 | 0.85 | 3.42 | 2.80 | 2.46 |
> | 2021    | 383 | 102 | 4.18 | 0.00 | 2.94 | 4.45 | 2.56 |
> | 2020    | 427 | 109 | 5.15 | 0.71 | 0.00 | 3.96 | 3.43 |
> | 2019    | 437 | 112 | 5.72 | 1.49 | 1.79 | 4.55 | 2.21 |
> | 2018    | 165 | 37  | 5.45 | 0.00 | 0.00 | 3.57 | 2.94 |
> | < 2018 | 89  | 36  | 4.49 | 0.00 | 2.78 | 3.37 | 1.09 |
>
> _Our original observation holds_:
> * Earlier years’ problems are not easier
> * Models trained on datasets w/ later cutoff dates != Better performance on more recent tasks (i.e. GPT-3.5 vs. GPT-4).
>
> The fact that different models are trained on data from different dates is an important issue for anyone dealing with evaluating LMs, and we believe that our analysis efforts show that our results are not tainted by this issue. SWE-bench’s ease of creating new issues via automatic collection also mitigate this concern for future users of this benchmark.
>
> 2. **SWE-Llama is not a significant contribution**: SWE-Llama is not the main contribution (full list in General Response). SWE-Llama is important because it establishes a lower bound for future attempts at the benchmark. It also shows the promise of fine-tuning on SWE-bench-train, which would likely improve performance for models like GPT-4.
>
> 3. **No comparison between CodeLlama and SWE-Llama**: As discussed in Section 4, we found that open source models like CodeLlama are completely unable to generate well formatted patches. CodeLlama’s performance is 0% in any retrieval setting, and patch generations consistently fail to apply. This prevents any meaningful evaluation of CodeLlama, and is the primary motivation for fine-tuning with SWE-bench-train. We thank you for bringing this up and have clarified this in the new version of our paper.
>
> 4. **Characterizing patches regardless of whether they applied or not**: Thanks for pointing this out. Based on your feedback, we recalculate the metrics (i.e. Total Lines, Added, Removed) for an average patch generation across all patches (not just successfully applied patches). Across all metrics + models, patch generations are much closer in size to gold edits. From this table, it’s clear that models struggle with generating longer patches that are correctly formatted. Further inspection of these flawed generations, as shown in Appendix F, indicate that hallucinations, abiding to existing code style/structure, and referencing long range dependencies correctly are common formatting errors that surface more frequently in longer generations.
>
> | Model | Total Lines | Added | Removed | Functions | Files |
> | --- | --- | --- | --- | --- | --- |
> | Claude 2 | 27.2 | 6.6 | 3.3 | 1.2 | 1.1 |
> | Claude 2 (Gold) | 61.6 | 17.8 | 8.6 | 2.6 | 1.4 |
> | ChatGPT-3.5 | 42.0 | 6.1 | 3.9 | 1.7 | 1.0 |
> | ChatGPT-3.5 Gold | 44.5 | 12.7 | 5.5 | 2.1 | 1.2 |
> | GPT-4 | 22.4 | 4.4 | 1.8 | 0.8 | 0.9 |
> | GPT-4 Gold | 50.3 | 14.0 | 6.5 | 2.3 | 1.3 |
> | SWE-Llama 13b | 68.9 | 9.5 | 4.3 | 2.5 | 1.6 |
> | SWE-Llama 13b Gold | 61.5 | 17.8 | 8.6 | 2.6 | 1.4 |
> | SWE-Llama 7b | 78.9 | 10.1 | 7.6 | 2.5 | 1.5 |
> | SWE-Llama 7b Gold | 65.1 | 18.8 | 9.0 | 2.7 | 1.5 |

---

> > ### Author Response · Authors · 2023-11-15
> >
> > 5. **Related Work**: Thanks for the suggestions, we will add these citations. While great inspiration, we note that evaluating these methods on SWE-bench is not possible because they are largely exploratory, do not deal with codebase-scale editing, and cannot solve SWE-bench.
> >
> > 6. **Performance on 25% Subset**: Based on your comment, we recompute the main evaluation results (% Resolved, % Apply metrics for BM25, Oracle retrieval settings) for the 25% subset of SWE-bench that GPT-4 was evaluated on to check if the results might be different. The performance here is consistent with the main results and conclusions already in the paper. This is expected because the selection procedure for the subset is random. We have added a discussion of this in the paper (Appendix C.2).
> >
> > | Model | BM25 Retrieval  | | "Oracle" Retrieval | |
> > |---|---|---|---|---|
> > || **% Resolved** | **% Apply** | **% Resolved** | **% Apply** |
> > | Claude 2    | 2.26 $\uparrow$ 0.30   | 28.57 $\downarrow$ 1.29 | 4.88 $\uparrow$ 0.08| 49.36 $\uparrow$ 2.37 |
> > | ChatGPT-3.5 | 0.17 $\downarrow$ 0.03 | 8.04 $\downarrow$ 2.46 | 0.84 $\uparrow$ 0.32| 11.67 $\downarrow$ 0.71 |
> > | GPT-4 | 0.00 $-$ 0.00| 4.50 $-$ 0.00    | 1.74 $-$ 0.00 | 13.24 $-$ 0.00 |
> > | SWE-Llama 7b| 0.42 $\downarrow$ 0.28 | 37.50 $\downarrow$ 0.34 | 2.12 $\downarrow$ 0.89 | 51.56 $\downarrow$ 3.24 |
> > | SWE-Llama 13b   | 0.70 $-$ 0.00| 40.47 $\uparrow$ 1.06 | 4.36 $\uparrow$ 0.39| 49.13 $\downarrow$ 3.01 |
> >
> > 7. **Selection of repositories for the benchmark**: Designing a unified approach to validation, installation, execution, and testing that works for many repositories is a complex technical challenge with many edge cases. Restricting to a single language simplifies this process.
> >
> >     We choose Python because it:
> >     * Is one of the most popular language on GitHub
> >     * Is the most popular language for code generation evaluation (e.g., HumanEval)
> >     * Has many high quality repositories (high # of stars, issues, PRs) maintained actively by open source contributors
> >
> >     Previous works have shown that results on Python coding benchmarks are highly correlated with results on other programming languages [1, 2].
> >
> > [1] Rozière, Baptiste, et al. ‘Code Llama: Open Foundation Models for Code’. arXiv [Cs.CL], 2023, http://arxiv.org/abs/2308.12950. arXiv.
> >
> > [2] Cassano, Federico, et al. ‘MultiPL-E: A Scalable and Extensible Approach to Benchmarking Neural Code Generation’. arXiv [Cs.LG], 2022, http://arxiv.org/abs/2208.08227. arXiv.

---

### Official Review · Reviewer_onWq · 2023-10-31

**Soundness:** 3 good
**Presentation:** 3 good
**Contribution:** 3 good
**Rating:** 6
**Confidence:** 4

**Summary:**

Authors aim to determine if LLMs can resolve real world software issues (vs constructing or fixing toy programs). Authors propose SWE-bench, a benchmark based on GitHub issues. They apply LLMs to try and fix these real-world issues and discover very poor performance.

**Strengths:**

- Authors present a good real-world problem benchmark based on real product sized GitHub repositories and real issues fixed in them.
- Fine tune CodeLlama 7B and 13B models to get at least somewhat positive performance on repository-wide code edits
- Propose retrieval methods to compose input for LLMs to fit into LLM context size.
- Evaluate LLMs on the benchmark and present general lessons from the results.

**Weaknesses:**

- Although benchmark and LLM evaluation on it are valuable, the paper does not present any novel solutions to the task in the benchmark. This limits the contribution.
- Please reorganize the paper so tables and figures are collocated with the text. Currently, it is hard to read when tables referenced out of order and explained very far from their location in the paper.

**Questions:**

This sentence, especially its last part, is unclear: "We compare the BM25 retrieval results against the oracle retrieval setting in Table 3, where we see that BM25 retrieves a superset of the oracle files in about 40% of instances with the 27,000 token context limit but only also excludes all of the oracle files in over half of instances.". I think this is trying to explain the results in Table 3 and trying to say that in around half cases BM25 does not retrieve any of oracle files. Is this what you are trying to say? Please explain or rephrase.

---

> ### Author Response · Authors · 2023-11-15
>
> Thanks so much for your time and effort towards this review, and for pointing out that SWE-bench is a good, real-world benchmark.
>
> We provide a list of updates in the general response comment above. Regarding your specific comments and suggestions:
>
> 1. **Lack of novel solutions to the task in the benchmark**: Past precedents such as the GLUE benchmark for NLU, WMT benchmarks for machine translation, and WikiText103 for LM perplexity, have shown the importance of simply presenting a tough benchmark for driving progress. We believe that presenting work which focuses primarily on introducing a challenging, useful benchmark can be very useful for the ML and NLP communities. Through experiments and analysis we also present actionable insights for future improvement (See Section 5, Appendix C/D/F).
>
> 2. **Reorganization of the tables/figures to be closer with text**: In the latest upload of our paper, we have incorporated this suggestion and put figures closer to text and made sure tables are mentioned in order. This should improve the readability. Thanks for this suggestion.
>
> 3. **Unclear sentence discussing retrieval**: Yes, your understanding is correct. To make the paper clearer we’ll be replacing the mentioned sentence with the following: "We compare the BM25 retrieval results with those of the 'oracle' retrieval setting, as shown in Table 3. We observe that in approximately 40% of instances, BM25 retrieves a superset of the oracle files for the 27000-token context limit. However, in almost half of the instances with the 27000-token limit, it retrieves none of the files from the 'oracle' context."
>
> Thank you again for your feedback and comments. If you have any remaining questions or concerns, we would be happy to address them.

---

> > ### Comment · Reviewer_onWq · 2023-11-18
> > **Thanks to the authors for comments and corrections**
> >
> > Thanks to the authors for their comments, clarifications and corrections

---

### Official Review · Reviewer_YqAB · 2023-10-31

**Soundness:** 3 good
**Presentation:** 4 excellent
**Contribution:** 4 excellent
**Rating:** 8
**Confidence:** 4

**Summary:**

The authors introduce a new benchmark and dataset for testing the abilities of LLMs to edit large code bases.  Previously existing test suites typically involve asking the LLM to generate a small self-contained function when given a natural language description.  In contrast, the new dataset requires the LLM to create a patch, which potentially affects many files across an entire repository, when given a bug report.

Bug reports and repositories were scraped from Github.  Ground truth is a human-written pull request, along with additional unit tests.  Success is determined by whether the patched repository passes additional unit tests that were supplied with the pull request.

The authors conduct numerous experiments with various LLMs, and discover that existing LLMs are (unsurprisingly) very bad at this task.  They analyze and discuss a number of issues as the cause of this failure, such as limited context length, difficulty in retrieving the relevant files from large datasets, poor test coverage, and the requirement that the model output a correctly-formatted patch, rather than ordinary code.

**Strengths:**

The primary contribution of this paper is the creation of a new dataset and methodology for evaluating the performance of LLMs on real-world software engineering tasks.  The benchmark is well-designed, and can be continually updated and expanded moving forward.  The experiments with existing models are interesting, but they mainly serve to illustrate that this is a difficult and unsolved problem.

I fully expect this to be a high-impact paper, because other practitioners working in this area can now measure the performance of their models against the new benchmark.  In addition, the analysis and discussion provided by the authors provides a good starting point for guiding future research in this area.

The qualitative analysis, which compares LLM-generated patches against human-generated patches was also quite insightful.

**Weaknesses:**

Generating a patch file, and generating code, are two very different tasks.  Existing models are pretrained on code, not patch files, so at least some of the poor performance could simply be due to the fact that the models are operating out of distribution on this data set.  (The authors mention this issue in the paper.)

**Questions:**

There is an additional issue with the way pretraining for code LLMs is typically done.  Due to context length limitations, the LLM often does not even see a complete file, much less a complete repository.   Moreover, the code fragments that are used for pretraining do not indicate what file they come from.

In contrast, in order to generate a good patch file, the model must be able to see the file and directory structure of the repository.  How do you handle file names and directory structure in your experiments?

---

> ### Author Response · Authors · 2023-11-15
>
> Thank you for your encouraging comments and suggestions.
>
> Please refer to the general response for a summary of our updates. Regarding your specific concerns and questions:
> 1. **Patch generation is out of domain**: We agree that LMs trained on code may seem out-of-domain for generating patch files. We ultimately think this is why open source models, including CodeLlama instruct and Llama 2 chat, are currently incapable of solving any issue in SWE-bench; they’re simply not on-par with proprietary models at instruction following yet.
>
>     We considered this problem as well in an ablation in Section 5, where we find for instance, when evaluated on the shortest half of inputs in the Oracle retrieval setting, Claude achieves 3.9% resolution rate when generating whole files compared to 7.8% when generating a patch.
>
>     _We therefore find that patch generation is both a practical and efficient modeling decision for our baselines, and that is why we decided to use it across the board_.
>
> 2. **Pretraining code doesn’t indicate the file names or paths**: You’re correct that pre-training data typically omits things like filenames, directory structures, etc. In all of our experiments, when representing code, we always wrap files’ contents with a start tag like [start of src/utils/readers.py] and an end tag [end of src/utils/readers.py], as well as prepend each line with its line number to help models create well formatted patch files. We’ve made this clearer in the description of our representations in the paper.
>
> We greatly appreciate your comments and suggestions. If you have any additional questions or concerns, please let us know. Thank you for your time and consideration.

---

### Official Review · Reviewer_r43j · 2023-11-03

**Soundness:** 3 good
**Presentation:** 4 excellent
**Contribution:** 3 good
**Rating:** 6
**Confidence:** 2

**Summary:**

The paper primarily describes a benchmark (Swe-Bench) for evaluating language models. The benchmark consists of issues reported in github python repositories. The authors give a detailed description of the criteria they used for constructing the benchmark. They also describe the inputs to the benchmark for evaluation. They finetune the CodeLlama model for the benchmark, and then evaluate this model and others using the benchmark.

**Strengths:**

The paper addresses a practically relevant issue, that of a benchmark for evaluating language models. The paper is clearly written, and quite a lot of work seems to have been done to support the material in the paper.

**Weaknesses:**

It seems that none of the models is doing well when the benchmark is used. It would be nice if the benchmark can be used to more clearly indicate where the problem in the language model lies. The results of the model evaluation e.g. difficulty correlates with context length or difficulty correlates with output length are expected and thus do not seem very interesting

**Questions:**

1) It would be nice if the exact contributions of the paper are stated more clearly.

2) In section1, the authors point out that there is a need for a challenging benchmark that can be used to check the abilities of language models. Although the results have been reported, I am not sure how far they evaluate the specific abilities or weaknesses. The results are general, and seem to apply to all the models without discerning the strengths/abilities or weaknesses of a particular model

3) At this stage, since all the models are performing poorly, perhaps there is a need for a benchmark that is neither too simple, but not as general as SWE-bench? Wouldn't this allow some aspects of the models to be better tested and reported?

---

> ### Author Response · Authors · 2023-11-15
>
> Thank you so much for your review.
>
> Please refer to the general response for a summary of our updates.
> To address your specific concerns and questions:
> 1. **More in-depth and insightful analysis**: Given the number and diversity of problems in SWE-bench, insightful analysis of generated solutions is certainly a challenge. To provide better insight, we’ve expanded our qualitative analysis in the paper to analyze 11 full generations from both Claude and SWE-Llama, showing that:
> Models tend to propose shortcut solutions which do not consider all possible inputs to the code.
> Models have a simplistic coding style that ignores the style and utilities of the overall codebase, leading to less accurate and portable solutions.
> We provide more insights and possible solutions, e.g., prompting with documentation, learning from interaction, and test generation, for solving these problems in the paper.
>
> 2. **Our exact contributions** — We restate our exact contributions in the general response comment and clarify them in the paper. Please refer to the general response for more information on this concern.
>
> 3. **Results are hard to interpret/distinguish between models**: We find that many of the challenges to solving long-context code problems are common to all models evaluated. While our results show that some models have better average success, overall performance on SWE-bench is so low across the board that it can be difficult to differentiate between models’ behavior. We think that SWE-bench can provide an important benchmark in guiding LM progress on the continual refinement of long context understanding, high and low-level reasoning, and grounded generation.
>
> 4. **Is there a need for a benchmark even simpler than SWE-bench?**: LM progress moves fast and evaluation benchmarks are quickly outdated. We believe that setting a bar as high as SWE-bench is of great interest to the community since it could serve as a more persistent aspirational achievement for LMs. Achieving high accuracy on SWE-bench would be both incredibly useful to actual software engineers and also indicate substantial technical progress in the development of LMs.
> Additionally, provided the structure of SWE-bench, it is possible for users to evaluate their models in slightly simpler settings (e.g. with the Oracle and Oracle-collapsed baseline), which can provide researchers some improved insights in the development of their own models.
>
> Thank you again for your feedback; we’ve uploaded a new version of the paper which addresses your concerns. If you have any remaining concerns or questions, please let us know.

---

### Author Response · Authors · 2023-11-15
**General Response**

We appreciate all reviewers’ time and great feedback! We have incorporated the rebuttal content into our revision to answer the reviewers’ questions and concerns.

SWE-bench’s motivation is simple: real world software engineering can serve as a challenging, and scalable testbed for evaluating the next generation of LMs. To show this, our contribution is multifold:
* **SWE-bench benchmark**: A challenging, high-quality, realistic, and easy-to-evaluate benchmark that introduces software engineering as a task. SWE-bench is an extremely useful goalpost for evaluating future LMs and automating software engineering.
* **Automatic Collection Pipeline**: SWE-bench is easily updatable with new issue-PR pairs from GitHub in case current LLMs memorize existing issues thanks to our carefully designed collection pipeline. Our pipeline can be pointed at new repositories to turn existing codebases into SWE-bench style testbeds.
* **SWE-Llama training data, code, and weights**: Open source models are entirely ineffective at the SWE-bench task. SWE-Llama is a critical proof-of-concept that shows the community that open source models can perform on par with proprietary SOTA models.
* **Experiments & Analysis**: We provide baseline performance and insights for various state of the art models. We perform ablations and show analysis related to model behavior identifying clear weaknesses and paths for future research.

In response to the reviewers’ questions and feedback, we have made several modifications
* **Clarified Language and Emphasized Contributions** (Reviewers onWq, r43j, YqAB): We updated language in the paper that reviewers found confusing, and changed our introduction to more clearly emphasize our major contributions as stated above.
* **More Qualitative Case Studies** (Reviewer r43j): We add five new case studies of Claude 2 (2 resolved, 3 unresolved) that provide deep insight into Claude. We also draw direct, qualitative comparisons between Claude and SWE-Llama (Appendix F).
* **New Analyses** (Reviewers onWq, r43j, BfZn): Evaluation on GPT-4 25% Subset (Appendix C.2), Extended Temporal Analysis (C.3), F2P + P2P Rate Analysis (C.4), Patch Gen. Extended Analysis (C.5), SWE Metrics (C.6)
* **Reorganized Tables + Citations** (Reviewer onWq): We have reordered tables in the main paper to be co-located with text references and added relevant works mentioned by the reviewers to our citations.

We thank the reviewers for their feedback, and we’re happy to answer any further questions.

---

### Meta-Review · Area_Chair_vVud · 2023-12-05

**Metareview:**

The paper introduces a new benchmark for language-driven code generation.  It tests whether LLMs can generate code to mimic real-world bugfix patches on real repositories.

Strengths:
- Large, realistic, challenging benchmark
- Achieving success on these problems will very likely produce interesting new research both on an applied and fundamental level.

Non-weaknesses:
- Existing models do poorly on the benchmark.  This is a strength: existing models solve around 80 of 2000 test cases.  This figure is enough to suggest the benchmark is not so hard as to be unreasonable, but hard enough to allow for plenty of innovation in this area.
- The paper doesn't propose new algorithms or models to solve the benchmark problem.  This is a dataset paper, it presents sensible baselines, and indeed should /not/ be trying to present new solutions to the problems.

Weaknesses:
- As reviewers point out, the evaluation requires that the model generate correctly-formatted patches, which is a somewhat stringent requirement - the models may do well on the "hard" problems of figuring out which code to edit, and what the fix should be, and then stumble on the tedious detail of formatting a patch.   The authors address this question in the rebuttal, showing that, for example, requiring the models to generate a full replacement code file lowers performance.
In the AC's view, this weakness is quite surmountable as future researchers target this benchmark - one might imagine approaches which e.g. use an intermediate representation of the patch that is forgiving of small errors, etc.

Additional suggestions:
- It might be of value to try to answer the question "how well could humans do on this task given the same prompts as the LLM"?

**Justification For Why Not Higher Score:**

n/a

**Justification For Why Not Lower Score:**

One might argue that "a dataset paper" does not need the widest exposure, but this dataset paper might inspire (a) research in a range of LLM-based techniques, and (b) other dataset papers in other domains.  Starting with such low baseline model performance might ultimately prove to be an error - the tests may simply not be realistic to solve, but is worth the experiment.

---

### Decision · Program_Chairs · 2024-01-16

Accept (oral)